# Beyond the Calorie Paradigm: Taking into Account in Practice the Balance of Fat and Carbohydrate Oxidation during Exercise?

**DOI:** 10.3390/nu14081605

**Published:** 2022-04-12

**Authors:** Jean-Frédéric Brun, Justine Myzia, Emmanuelle Varlet-Marie, Eric Raynaud de Mauverger, Jacques Mercier

**Affiliations:** 1PHYMEDEXP, Université de Montpellier, CNRS, INSERM, CHU de Montpellier, 34295 Montpellier, France; justine.myzia@gmail.com (J.M.); eric.raynaud-de-mauverger@chu-montpellier.fr (E.R.d.M.); jacques.mercier@umontpellier.fr (J.M.); 2IBMM, Université de Montpellier, CNRS, CHU de Montpellier, 34090 Montpellier, France; emmanuelle.varlet@umontpellier.fr

**Keywords:** calorimetry, LIPOXmax, FATmax, training, lipid oxidation, fat, maximal fat oxidation (MFO), fat metabolism, indirect calorimetry, peak fat oxidation, substrate oxidation, carbohydrate oxidation, sleeve gastrectomy, myometabokines

## Abstract

Recent literature shows that exercise is not simply a way to generate a calorie deficit as an add-on to restrictive diets but exerts powerful additional biological effects via its impact on mitochondrial function, the release of chemical messengers induced by muscular activity, and its ability to reverse epigenetic alterations. This review aims to summarize the current literature dealing with the hypothesis that some of these effects of exercise unexplained by an energy deficit are related to the balance of substrates used as fuel by the exercising muscle. This balance of substrates can be measured with reliable techniques, which provide information about metabolic disturbances associated with sedentarity and obesity, as well as adaptations of fuel metabolism in trained individuals. The exercise intensity that elicits maximal oxidation of lipids, termed LIPOXmax, FATOXmax, or FATmax, provides a marker of the mitochondrial ability to oxidize fatty acids and predicts how much fat will be oxidized over 45–60 min of low- to moderate-intensity training performed at the corresponding intensity. LIPOXmax is a reproducible parameter that can be modified by many physiological and lifestyle influences (exercise, diet, gender, age, hormones such as catecholamines, and the growth hormone-Insulin-like growth factor I axis). Individuals told to select an exercise intensity to maintain for 45 min or more spontaneously select a level close to this intensity. There is increasing evidence that training targeted at this level is efficient for reducing fat mass, sparing muscle mass, increasing the ability to oxidize lipids during exercise, lowering blood pressure and low-grade inflammation, improving insulin secretion and insulin sensitivity, reducing blood glucose and HbA_1c_ in type 2 diabetes, and decreasing the circulating cholesterol level. Training protocols based on this concept are easy to implement and accept in very sedentary patients and have shown an unexpected efficacy over the long term. They also represent a useful add-on to bariatric surgery in order to maintain and improve its weight-lowering effect. Additional studies are required to confirm and more precisely analyze the determinants of LIPOXmax and the long-term effects of training at this level on body composition, metabolism, and health.

## 1. Introduction: Can Exercise Be Defined on the Basis of the Fuel It Oxidizes?

Around the year 2000, several teams [1,2,3] proposed to categorize exercise based on the energetic substrate that is used as a source of energy by muscle. The literature of the preceding decade had shown that carbohydrates and lipids are the two major fuels oxidized by the exercising muscle [4,5,6] and that lipid oxidation reaches a maximum (maximal fat oxidation, MFO, or peak fat oxidation, PFO) at a variable level grossly between 40 and 50% of the maximal aerobic capacity. Above this level, the percentage of energy provided by carbohydrate oxidation increases, and carbohydrates become the predominant fuel.

The intensity level at which MFO occurs is generally not a precise value of exercise intensity but rather a zone (lipid oxidation zone). Its upper limit has been termed, according to the teams that used different methods, LIPOXmax [1], FATOXmax [2], or FATmax [3]. Most authors prefer to write “FATmax” although this refers to a specific technique using ultra-short 3 min steady-state steps that is not beyond criticism. In this review, we will use “LIPOXmax”.

Attempts to train obese subjects at this level, in order to help them to lose weight, were then proposed [7,8,9], already showing promising results. However, this new approach to exercise training in obesity (and other metabolic diseases) did not reach a great audience and the majority of exercise studies, over the following 20 years, almost always referred to a percentage of the maximal aerobic capacity rather than lipid or carbohydrate (CHO) oxidation [10,11]. Apparently, since a dose–response relationship between the volume of aerobic exercise and fat reduction was known to exist [12], it seemed unlikely to many specialists that such a minimal increase in energy expenditure would result in any significant biological effect.

More recently, however, a growing interest in this concept developed two decades ago has emerged [13,14,15]. A series of interesting new findings that were published during this period may explain this. First, the weight-lowering effect of this approach was fairly confirmed by well-conducted meta-analyses [16]. Furthermore, this weigh-lowering effect was unexpectedly shown to be very prolonged, over several years, resulting in a slow, sustained loss of fat mass [17,18]. The apparent paradox of the efficiency of this low-intensity, low-volume training strategy found some likely explanations. The series of studies (“STRRIDE”, i.e., “Studies of a Targeted Risk Reduction Intervention through Defined Exercise”) have elegantly demonstrated that low-intensity, low-volume exercise training has even more beneficial effects on fat deposits and insulin sensitivity than higher exercise levels [19,20]. This series of studies emphasize the fact that even a modest amount of regular exercise is able to counteract obesity evolution and metabolic deterioration [21] since skeletal muscle metabolites, such as succinate, are released, associated with improved insulin sensitivity, and likely play a role in this mechanism [22]. Exercise has also been shown to modify the epigenetic state and therefore induce transcriptional changes via transient modifications in the bioavailability of metabolites, substrates, and cofactors. Among these mechanisms, lipid handling by the oxidative metabolism, as well as the activation of the Tricarboxylic Acid Cycle, has been hypothesized to play a role [23]. More precisely, lipid breakdown can generate short-chain molecules such as butyrate, which have been demonstrated to regulate histone deacetylase [24]. Exercising muscle can also release bioactive substances known as myokines that can exert beneficial actions at the whole-body level [25]. In addition, Tricarboxylic Acid Cycle intermediates (citrate, α-ketoglutarate, fumarate, and succinate) seem to play an important role and have been recently called myometabokines [26]. They are released, presumably as well as other active molecules such as β-aminoisobutyric acid, which appears to promote adipose tissue browning, increase insulin sensitivity, and protect against obesity induced by a high-fat diet [27]. A bout of low-intensity exercise, when performed at the beginning of the day, increases lipid oxidation over the following 24 h [28,29]. Finally, low-intensity exercise appears to regulate eating behavior and reduce sedentarity-induced overeating [30]. Therefore, exercise can no longer be considered simply a means to waste energy. Clearly, it has additional important effects that may explain why it is more powerful than could be expected from a calorie deficit alone [31].

For all these reasons, as investigators involved since the beginning of this research [32] and still working on this topic [33], we thought that it was necessary to propose a new comprehensive review on this issue, updating previous reviews conducted by our team [34,35,36,37] and others [9,13,14] and considering the new information obtained since then. The research publications reviewed in the present study were all obtained from Pubmed (https://pubmed.ncbi.nlm.nih.gov/) (accessed on 17 January 2022), Archive ouverte HAL (https://hal.archives-ouvertes.fr/) (accessed on 17 January 2022), and science direct (https://www.sciencedirect.com/) (accessed on 17 January 2022).

## 2. The Balance of Substrates during Exercise

### 2.1. Balance of Substrate Oxidation during Exercise: The “Crossover Concept”

It has been known since the beginning of the 20th century that carbohydrates and fats are the major sources of energy for muscle, both at rest and during exercise [38,39,40]. Obviously, proteins are also a significant fuel during exercise [41] but their contribution to the energy supply during exercise is less important. Carbohydrates can be considered the major fuel, but intramuscular and circulating lipids also represent an important source of energy during exercise [5,6,42]. Figure 1 summarizes the “crossover concept”, which was proposed for describing the shift in the balance of substrates that occurs around 50% of the maximal aerobic capacity [43].

### 2.2. Measuring the Maximal Lipid Oxidation during Exercise

After the publication of the “crossover concept” [43], several teams developed an exercise test assessing the balance of substrates. The first paper by our team in 2001 [32] was followed in 2002 by two more on the same topic by other teams [2,3] who used a slightly different methodology, as discussed below. 

Based on previous studies on calorimetry during long-duration steady-state workloads [44,45], Perez-Martin and coworkers published a protocol [32] consisting of five 6-min submaximal steps. The duration of 6 min was chosen in order to obtain a steady state for gas exchanges. Since there is no need to have a precise evaluation of VO_2_peak during exercise calorimetry, the maximal level can be indirectly evaluated by linear extrapolation of VO_2_ until the theoretical maximal heart rate according to the ACSM guidelines (VO_2max_ ACSM) [46]. 

The test is performed on an ergometric bicycle and includes continuous measurement of VO_2_ and VCO_2_ and EKG monitoring. The workloads are set at approximately 30, 40, 50, and 60% of predicted maximal power, but these levels can be adapted to the patient during the test according to the respiratory exchanges ratio (RER), in order to obtain values of RER below and above 0.9, which is the RER at the point of crossover, and also a value above 1 indicating the level where the subjects are no longer oxidizing fat (LIPOX zero or “FATmin”, see below). The values of VO_2_ and VCO_2_ measured during the fifth and sixth minutes of each stage are assumed to represent a steady state and thus used for the calculation with the equations of indirect calorimetry [47,48,49,50]:Carbohydrates (mg/min) = 4.585 VCO_2_ − 3.2255 VO_2_(1)
Lipid Oxidation (mg/min) = −1.7012 VCO_2_ + 1.6946 VO_2_(2)

The rationale for using 6 min steps and performing calculations on the values of the 5–6th minute is based on a study by McRae and coworkers [51], which indicates that, at this time, the CO_2_ production from bicarbonate buffers becomes negligible. There has been controversy over the best duration of the steps and many authors prefer to use shorter steps [3]. We have reported [52] that 3 min steps result in a systematic overestimation of MFO. In addition, some reports show poor reproducibility of methods employing 3 min steps [53,54,55] or 4 min steps [56]. 

Nevertheless, these various protocols provide the same information. They show that the increase in lipid oxidation displays an asymmetrical dome-shaped curve. This curve culminates at the level of MFO at an intensity termed, in this protocol, the LIPOXmax, and then lipid oxidation decreases at higher power intensities. The power intensity where it becomes equal to zero is the point where RER is equal to 1 and is termed the LIPOXzero (or FATmin).

The empirical formula (Equation (2)) that provides the lipid oxidation rate is, as mentioned above:Lipid oxidation (mg/min) = −1.7 VCO_2_ + 1.7 VO_2_(3)

It is easy to deduce from this formula that the relation between power (P) and the oxidation of lipids (Lox) displays an asymmetrical dome-shaped curve of the form:Lox = A·P (1 − RER)(4)

Derivation of this curve enables us to calculate the power intensity at which lipid oxidation becomes maximal, which is the point where the derivative becomes equal to zero. Therefore, the LIPOXmax calculation is only an application of the equation of lipid oxidation used in calorimetry and is model-independent [36]. Other authors proposed visual determination of this level on the curve [2,3], a sophisticated model using sinus calculation (termed “sine model, or SIN”) [57], or a polynomial-curve analysis [58]. There is no consensus about the best technique, and, overall, they all appear to provide almost the same information, but as discussed below, they do not have the same reproducibility, and perhaps the choice among them could be based on this aspect. It should be pointed out that most of the time, the calculation of a very precise top of the curve is somewhat theoretical and that it would be better to speak about a “lipid oxidation zone” or “LIPOX zone” whose width is variable among subjects (see an example on Figure 2). This issue is discussed in Achten and Jeukendrup’s pioneering study [3] in which the authors define a “Fat(max) zone” (see also Figure 3), which is found, in moderately trained cyclists, to cover a rather large range of intensities between 55 ± 3 and 72 ± 4% VO_2max_ and to rapidly fall above the LIPOXmax.

Maximal fat oxidation (MFO) can be expressed as crude results of flow rate (mg/min) or corrected for fat-free mass [2,61] or muscle mass [62].

### 2.3. Reproducibility and Stability over Time

Initial studies on exercise calorimetry unanimously reported fair reliability, which seems to be confirmed by daily clinical practice. The coefficient of variation for the LIPOXmax (at that time, it was manually determined) was found to be 11.4% [32], and with Achten and Jeukendrup’s procedure, it was 9.6% [63]. Michallet and coworkers found a CV equal to 8.7% with the LIPOXmax procedure [64]. However, there has been some controversy about this reproducibility. When LIPOXmax is assessed with 6 min steps in well-standardized conditions via the derivation of the equation of lipid oxidation, it has been shown by Gmada and coworkers [65] to result in a CV as low as 5.02%. More recently, De Souza Silveira and coworkers also found, for a test using stages of 6 min on a treadmill, a CV of 5% for the velocity at which fat oxidation is maximal, and 7.0% for an exercise intensity eliciting peak fat oxidation [66]. Studies using less-standardized conditions of testing for fasting and rest found a wider variability [53]. Croci and coworkers [54] reported poor reproducibility of the LIPOXmax measured with 5-min stages, either with crudely measured values, a third polynomial curve, or the sophisticated sine modelling, which yielded CVs of 18.6, 20.8, and 16.4%, respectively. Similarly, in a large cohort of 99 healthy subjects, Chrzanowski-Smith and coworkers found, for protocols using 5- and 2-min steps, a within-subject coefficient of variation of 21% for MFO and 26% for FATmax expressed as %VO_2max_ [67]. Considering that 3-min stages were perhaps insufficient to attain a steady state, the same authors investigated 4-min stages and evidenced a wide intraindividual variation in PFO [56]. However, using the LIPOXmax protocol, Marzouki and coworkers confirmed that, if the conditions of measurement are well standardized, crossover point, LIPOXmax, and MFO measurements are reproducible, with coefficients of variation of 9.0, 2.6, and 15.2%, respectively [68].

This issue is carefully discussed by Maunder and coworkers [14] who conclude, in agreement with our previous reports [36], that the measurement of the level of maximal lipid oxidation, either with the LIPOXmax or the FATmax procedure, is reproducible if performed in standardized conditions, but that it can be modified by various situations. 

In addition, we reported that there is remarkable stability with the LIPOXmax, which remains stable over a period of 30 months if diet and exercise habits are not modified [69].

### 2.4. Measuring the Kinetics of Carbohydrate Oxidation during Exercise

The increment in CHO oxidation above basal values is known to be exponential [70,71], but over the range of intensities used during graded exercise calorimetry, it displays an almost linear curve as a function of the developed power, as can be seen in Figure 2. The slope of this relationship is the *carbohydrate cost of the watt* [72,73]. 

## 3. Physiological Relevance and Determinants of LIPOXmax

In most studies, the measurement of MFO was shown to predict the fat oxidation rate at a steady state at the LIPOXmax for 40–60 min, before fat oxidation increases due to the glycogen depletion [74,75]. However, recently, Özdemir and coworkers reported that in some sedentary individuals, fat oxidation rates may gradually decrease, resulting in a lower oxidation rate than expected after 16 min [76].

### 3.1. The LIPOXmax Is the Level of Exercise Spontaneously Selected for Prolonged Exercise: The Hypothesis of the ‘Healthy Primitive Lifestyle’

Therefore, the level of maximal lipid oxidation is a biological characteristic of an individual, which remains rather stable, although it can be modified by many factors. There are some other arguments suggesting that this parameter has physiological relevance.

Dasilva was the first to observe that when people were asked to select a pace to maintain for a 1-h exercise, they spontaneously selected an intensity within the lipid oxidation zone [77]. In another work, we reiterated their observation [78].

This point, together with the other properties of the LIPOXmax that we review in this paper, prompted us to propose the unifying concept of the «healthy primitive lifestyle» [79].

According to this theory, this level of activity, which spares carbohydrate stores, uses fat stores, and minimizes the orexigenic drive, was probably used several hours per day by our Paleolithic hunter–gatherer ancestors, during the period where polymorphisms coding for the “thrifty genotype” were selected by evolution. The concept of the “thrifty genotype” has been proposed by James V. Neel in 1962 [80] and assumes that genetic variants promoting fat storage during periods of food abundance were evolutionarily advantageous and thus were positively selected. Food abundance and physical inactivity eliminate the benefits of this genetic profile and induce metabolic derangements leading to obesity and Type 2 diabetes. In this context, it has been proposed that some “physical activity genes” whose expression requires a quantity of muscular activity above a certain threshold are the “thrifty genes” of our hunter–gatherer ancestors. In modern life, due to the lack of physical activity, these ancestral genotypes result in increased storage of lipids [81]. Archeologic evidence suggests that our Paleolithic ancestors were obviously able to perform, when necessary, high-intensity exercise in situations of hunting or conflict, but most of the time exercised at a low to moderate intensity, i.e., at the level where the muscle mostly oxidizes lipids [82]. Insufficient volumes of such physical activity favor the excess storage of fat. This storage of fat has been shown to improve the ability to survive a pathologic event associated with temporary immobilization (obesity paradox [83]), but of course, obesity is deleterious, even when it is termed “healthy” because it seems to be not associated with metabolic or cardiovascular disorders [84]. Training profiles corresponding to high-level sports are associated with a dramatic reduction in fat mass, but these profiles of “warriors” were probably found only in a subset of the population. Between these two situations on the edge of physiology, exercise at the LIPOXmax, which was regularly practiced by the majority of people, fit with the thrifty genotype, moderating food intake, hypoglycemic cravings, and fat mass. Food moderately enriched in proteins reinforces this strategy by increasing lipid oxidation during exercise. This model explains the interest in the metabolic syndrome of LIPOXmax exercise associated with a diet moderately enriched in protein (1.2 g·kg^−1^·d^−1^), with low-glycemic-index carbohydrates and low in fat. Such a lifestyle is adapted to the “thrifty genotype” inherited from our Paleolithic ancestors because of its effects on energy metabolism and eating behavior. This concept also provides an explanation of some of the unexpected findings on relationships between rheological properties of blood and exercise in sedentary subjects and athletes [85]. 

### 3.2. The Balance of Substrates as a Window on Mitochondrial Function and Metabolic Flexibility

In studies of muscle biopsies, it was observed that measurements of lipid and CHO oxidation during exercise are correlated to mitochondrial function. This was first reported in diabetics [86] in whom LIPOXmax is correlated to citrate synthase activity, and in whom the improvement in mitochondrial respiration after training is correlated to an increase in MFO. Nordby and coworkers [87] did not find these relationships, which were evidenced by Rosenkilde and coworkers [88]. These authors found that exercise training increased MFO parallel to muscle expression of citrate synthase, β-hydroxyacyl-CoA dehydrogenase, and mitochondrial OXPHOS complexes II–V, but the multivariate analysis did not select these mitochondrial parameters as determinants of MFO, suggesting that they are not very close determinants of it. These measurements were predictors of fat oxidation during prolonged fed-state cycling [89]. Sahlin and coworkers also clearly demonstrated that whole-body fat oxidation during low-intensity exercise (35% VO_2max_) is correlated with that measured in vitro in isolated mitochondria. The correlation was observed during exercise at 80 and 120 W and was even stronger when interpolated to the same relative intensity, i.e., fatty oxidation potential at the mitochondrial level influences whole-body fat oxidation during low-intensity exercise [90]. More recently, Maunder and coworkers [83] observed, in athletes, correlations between MFO, vastus lateralis citrate synthase activity, and abundance in the fatty acid translocase CD36. 

In contrast, when exercise training is performed at a high intensity, mitochondrial markers of CHO processing rather than lipid processing are improved, evidencing, once again, that the balance of substrates is a reflection of mitochondrial function [91].

The ability of the mitochondria to switch from fat to carbohydrate oxidation in response to changing physiological conditions and therefore to adapt fuel oxidation to fuel availability has been termed metabolic flexibility [92], and its defect, termed metabolic inflexibility, is observed in insulin resistance and type 2 diabetes [93,94]. Lifestyle changes in dietary fat intake, physical activity, and weight loss may help to reverse metabolic inflexibility in skeletal muscle, and thereby contribute to the prevention of type 2 diabetes [95]. Since metabolic inflexibility in the metabolic syndrome is characterized by both a decreased capacity to oxidize lipids and an early transition from fat to carbohydrate oxidation associated with an elevated blood lactate concentration as exercise intensity increases, San-Millán and Brooks [96] recently proposed the assessment of metabolic flexibility with the measurement of lipid oxidation and blood lactate kinetics during exercise.

### 3.3. Carbohydrate Breakdown Controls Lipid Oxidation Rate

The cellular mechanisms underlying the “asymmetrical dome-shaped curve” of lipid oxidation have been reviewed by Sahlin and coworkers [42]. We summarize them in Figure 3. Since lipid oxidation decreases even if additional fat is provided, lipid oxidation is not limited by lipid supply. As shown by Watt and coworkers [60], hormone-sensitive lipase activity, which regulates intramuscular triacylglycerol hydrolysis in skeletal muscle, is increased to an even greater extent during high-intensity exercise (90%) compared to low- and moderate-intensity exercises (30% and 60%, respectively), while at this level, lipid oxidation is blunted. Similarly, lipolysis in the adipose tissue exhibits a gradual increase proportional to the increase in exercise intensity between 20 and 70% VO_2max_ [97]. Accordingly, circulating values of non-esterified fatty acid and mean glycerol are still increasing at the highest levels of an incremental exercise test (60% VO_2max_) while lipid oxidation has vanished [98]. Above this level, despite high lipolytic rates in peripheral adipose tissue, the rate of release of free fatty acids into plasma progressively declines with increased exercise intensity so that plasma-free fatty acids concentrations decrease at 85% VO_2max_ [4]. Therefore, lipolysis is not the limiting factor for lipid oxidation during exercise. The limiting steps seem to be downstream lipolysis, at the level of the entrance in mitochondria, governed by CPT-1, which can be inhibited by Malonyl-CoA and lactate [99], and possibly downstream CPT-1 other mitochondrial enzymes such as Acyl-CoA synthase and electron transport chain. These steps are limiting for fat oxidation because they are sensitive to the rate of CHO oxidation, so an increase in CHO oxidation results in a decrease in lipid oxidation. Experiments using intravenous infusion of labeled long-chain fatty acids during 40 min of cycling at a steady state at 50% of VO_2max_ provide clear evidence that carbohydrate availability directly down-regulates lipid oxidation during muscular activity. In this case, there is an increase in glycolytic flux, and this increased flux blunts long-chain fatty acid oxidation [100]. Conversely, it is important to mention that a decrease in carbohydrate availability, associated with glycogen depletion, reverses this inhibition and is therefore associated with an increase in lipid oxidation.

### 3.4. Agreement between the Balance of Substrates and Lactate/Ventilatory Threshold

The crossover concept [43] implies that there is a critical intensity level of exercise where there is a metabolic shift, at approximately 50% of VO_2max_, and it was thus logical to assume that the crossover point (COP), the peak of lipid oxidation, the point where blood lactate rises, and the ventilatory threshold (V_T_) were closely related. Agreement between COP and V_T_ was reported by Aissa Benhaddad and coworkers [101] and Ceugniet and coworkers [102], while Astorino and coworkers [103] showed in moderately trained young women that V_T_ and LIPOXmax occurred at the same level. In 56 trained male athletes, Perric and coworkers report that V_T_ and LIPOXmax occur at approximately the same level, 45.95 ± 10.21% and 47.47 ± 10.59% of VO_2max_, respectively, and are highly correlated [104]. Even more recently, Emerenziani and coworkers investigated this question again and concluded that V_T_ occurs at a slightly higher %VO_2peak_ (grossly + 3%) than the LIPOXmax and that V_T_ and LIPOXmax are well correlated [105].

Two studies showed that lactate increases in the blood above the LIPOXmax [59,106], consistent with the report that both the COP and the LIPOXmax are negatively correlated to blood lactate concentrations [107]. These results are in agreement with the idea that the increase in blood lactate is due to an increased glycolytic flux, elicited by an increased work rate [108], which, at the same time, inhibits lipid oxidation [100].

### 3.5. The LIPOXmax as a Marker of Functional Capacity

Maximal fat oxidation in obese children is a statistical determinant of the six-minute walking test, so a predictive equation for the LIPOXmax can be developed from this measurement [109]. This further indicates that the LIPOXmax is one of the measurements, among others, that provide information about an individual’s ability to exercise. Furthermore, Maunder and coworkers reported that including MFO in a predictive equation of endurance could explain an additional ~2.6% of the variation in performance (in addition to peak oxygen uptake, power at 4 mmol·L^−1^ blood lactate concentration, and gross efficiency). All this suggests that the assessment of lipid oxidation has clear relevance to sports medicine [89].

## 4. Various Influences

Comprehensive updates about mechanisms that determine the maximal level of lipid oxidation have been recently published [14,110], pointing out the importance of training status, intensity, duration, sex differences, and nutrition. These reviews gather a great deal of important information, and their content will not be fully recapitulated here. However, in the following paragraphs, we will add some important points (Table 1). 

In a recent synthesis on 5258 exercise calorimetries, we found that the point of maximum lipid oxidation is 47.1% VO_2max_ (Figure 4). At this level, subjects oxidize, on average, 209.5 mg/min of fat. These parameters are widely variable among subjects, although they remain stable over time for a given individual if there is no change in lifestyle. In fact, the LIPOXmax is significantly shifted to the right by 8 to 15% in women compared to men. When lipid oxidation is expressed per unit of muscle mass, women oxidize more lipids (+10%) than men. Lipid oxidation during exercise also significantly decreases with age. This large database confirms that the ability to oxidize lipids to exercise is slightly higher on average in women and is decreased with obesity and aging [153].

### 4.1. Genetics

Perhaps the most important information that has recently emerged is the role of genetics, which was not yet evidenced. In a study on 23 male monozygotic twin pairs, Karppinen and coworkers [156] found that lipid oxidation rates were correlated within co-twins, both at rest and during exercise, demonstrating that genetics is a major determinant of exercise-induced fat oxidation. In addition, MFO was correlated with the past 12-month leisure-time physical activity and the Baecke activity score, confirming that physical activity also markedly influences this parameter. Therefore, hereditary factors appear to be even stronger determinants of fat oxidation (at rest and during exercise) than leisure-time physical activity. Besides, MFO was negatively correlated with the area under the curve of insulin and glucose during an oral glucose tolerance test, showing that peak fat oxidation is associated with better metabolic health while resting fat oxidation does not exhibit this correlation. This study confirms the concept that lipid oxidation during exercise is a marker and/or determinant of metabolic health, which is, in turn, well known to be under the influence of both genetic and lifestyle factors [157].

### 4.2. Effects of Exercise and Training on Lipid Oxidation

Both muscular activity and sedentary time modify MFO and LIPOXmax [142]. Exercise stimulates lipid oxidation in at least five circumstances: 

(a) Exercising muscle may oxidize fat as a source of energy during a steady-state exercise, and the rate of this oxidation is *intensity-dependent*, i.e., it displays an asymmetrical dome-shaped curve as a function of the intensity level [43]. This issue is the main topic of the current review. 

(b) This rate of fat oxidation can be *time-dependent*, i.e., it increases when exercise is heavy and prolonged, resulting in significant glycogen depletion [158,159,160,161,162]. This time-dependent increase in lipid oxidation remains negligible during low- to medium-intensity exercise as long as the duration of this exercise does not exceed one hour. In this case, the lipid oxidation rate has been shown to display a plateau [75,163].

(c) The rate of lipid oxidation can also increase after high-intensity exercise [164,165] (i.e., a variety of exercises that oxidize almost exclusively CHO) [160], but this postexercise lipid oxidation after a high-intensity bout is often rather moderate [166,167]. This postexercise lipid oxidation is proportionally higher in men than women [168]. It may represent a mechanism for sparing CHO and facilitating postexercise glycogen repletion [169]. Overall, this issue of postexercise increased lipid oxidation, which generated some controversy fifteen years ago, likely has little practical relevance and we will not further develop it in this review. According to Warren and coworkers [170], the most appropriate strategy to burn fat during exercise is to perform this exercise in the LIPOX zone, i.e., at a low to medium intensity level.

(d) Long-term regular exercise can enhance fat oxidation at rest, thus shifting the balance of substrates oxidized over 24 h toward oxidation of higher quantities of lipids [171]. This issue is potentially quite interesting, because a training-induced increase in the ability to oxidize lipids over 24 h is statistically a predictor of exercise-induced weight loss [172]. Therefore, exercise strategies aiming at increasing 24 h lipid oxidation are likely to be useful in the management of obesity. The definition of the types of exercise that increase 24 h lipid oxidation is beyond the scope of the current review.

(e) As already mentioned above, another type of exercise-induced increase in lipid oxidation has been more recently evidenced and may have great potential in the management of obesity. It has been shown that a 60 min bout of endurance exercise at 50% VO_2max_ performed before breakfast increased fat oxidation over 24 h. The same exercise in the afternoon or the evening did not have this effect. This increase in lipid oxidation over 24 h was negatively correlated with the transient deficit in energy, and more precisely in carbohydrates, induced by this morning exercise [28,29]. In another study, these authors reported that the increase in the 24 h fat oxidation was probably induced by a greater rise in the plasma-free fatty acid concentration and unsaturated/saturated ratio compared to what occurs when exercise is performed in a postprandial state. This increase in plasma-free fatty acids was related to the transient carbohydrate deficit after exercise [173]. These findings confer a new interest in the issue of exercise performed while fasting before breakfast, which was, until now, mostly analyzed in terms of crude energy deficit and hunger [174].

(f) Previous exercise increases MFO

Consistent with the abovementioned observation that CHO oxidation blunts fat oxidation during exercise and that this effect is reversed when the CHO supply to muscles is decreased, previous exercise of 1 h has been shown to result in an increase in MFO [119]. Interestingly, this effect is no longer found after 2 days of fasting [150].

(g) Training

As will be developed later, low-intensity exercise training in various pathologies is able to improve the balance of substrates, increasing MFO and shifting LIPOXmax to the right. 

In young sedentary overweight men, Rosenkilde and coworkers [88] showed that both high-dose (600 kcal/day) and moderate-dose (300 kcal/day) exercise training increased MFO together with skeletal muscle expression of citrate synthase, β-hydroxyacyl-CoA dehydrogenase, and mitochondrial oxphos complexes II–V. 

Trained athletes therefore also exhibit particular patterns of the balance of substrates with some of them prone to endurance sports who were lipid oxidizers, and others who relied on carbohydrates and were mostly able to perform high-intensity, short-duration exercise [175]. Therefore, it could be interesting to analyze, among other parameters, the ability of athletes to oxidize fat, which may represent a metabolic advantage for prolonged exercise (“metabolic endurance”) [36]. In this respect, a large recent study on exercise calorimetry in 1121 athletes representing a wide variety of sports and competitive levels shows values of MFO ranging from 0.17 to 1.27 g/min (on average, 0.59 ± 0.18 g/min) and occurring at an exercise intensity ranging from 22.6 to 88.8% (on average, 49.3 ± 14.8%) of VO_2max_ [61,176]. Coquart and coworkers reported LIPOXmax values in trained cyclists occurring at quite various levels in the range of 41–47% VO_2max_ [177]. 

(h) Running vs. cycling

It has been shown since the earliest studies that MFO is higher during running on a treadmill than during cycling [63], and this issue was more recently investigated by Chenevière and coworkers [178] who reported via the SIN procedure that MFO occurs at a significantly higher %VO_2max_ (57.2 ± 1.5% vs. 44.2 ± 2.9%). Presumably, a greater muscle mass is involved in running than cycling, explaining most of these differences. These differences have also been found by other authors [179], but they did not find such a wide range in averages [180].

### 4.3. Age and Gender

Since the first studies at the beginning of this century, age and gender were shown to influence MFO and LIPOXmax [136]. In a study on 304 subjects [128], we compared exercise calorimetry results among athletes, sedentary, and type 2 diabetics, in whom men and women were matched for age, BMI, and physical activity, and we found that the LIPOXmax occurs at a higher percentage of VO_2max_ in female athletes and sedentary women compared to men. However, MFO expressed for fat-free mass or muscle mass did not differ between men and women, as recently confirmed by Frandsen and coworkers [150]. In the review by Maunder and coworkers [14], gender, but not age, is considered a major determinant of the LIPOXmax, while Amaro-Gahete and coworkers [181] clearly confirm the influence of gender but indicate that participants’ age should be considered when providing normative values. In our already-mentioned personal series of 5258 exercise calorimetries, we found that the LIPOXmax is significantly shifted to the right by 8 to 15% in women compared to men. When MFO is related to muscle mass, women oxidize significantly more lipids (+10–21%) than men. Lipid oxidation during exercise also significantly decreases with age [153]. In Randell and coworkers’ study of 1121 athletes, MFO was higher in male athletes compared to females when expressed as crude values (0.61 and 0.50 g·min) and slightly higher in females when expressed as relative to fat-free mass [61,176]. On the whole, the difference between men and women for lipid oxidation does exist, but it is not so marked, since it requires large series to be evidenced on values normalized for anthropometry and VO_2max_.

However, this difference may have some physiological relevance as suggested by the report by Dasilva and coworkers [77], as in individuals exercising at a self-selected pace, fat oxidation contributed more to total energy expenditure in women than men, even if both genders selected a similar exercise intensity.

By contrast, throughout the course of puberty, MFO markedly decreases and the LIPOXmax is progressively shifted toward lower intensities [120,121]. This is also found in obese teenagers [182] in whom exercise training should therefore be targeted at a lower intensity, even more since obesity also shifts their ability to oxidize fat at a lower intensity [183].

### 4.4. Feeding and Dietary Habits

One of the more classical factors that modify the balance of substrates during exercise is nutrition (Figure 5). A previous meal exerts an acute effect on the curve of lipid oxidation. Taken less than 3 h before exercise, a meal containing carbohydrates both decreases MFO and shifts LIPOXmax to a slightly lower intensity [5,6,111,112,113].

Nutrition is also likely to exert a chronic effect. Nutritional habits, more precisely dietary carbohydrate intake, appear to be weak but significant statistical predictors of the interindividual variability of the capacity to oxidize fat during exercise [114].

The influence of some polyphenols has been described. For example, decaffeinated green tea extract has been reported to dramatically increase (+45%) MFO and shift the point where lipids are no longer oxidized (termed LIPOX zero or FATmin) toward intensities higher than 22.5% [115]. This effect reported during exercise involving the legs is no longer found during arm cycle exercise [116]. By contrast, another source of polyphenols, Montmorency tart cherries (*Prunus cerasus* L.), which are rich in anthocyanins and are reported to increase fat oxidation, has been found to exert no effect on the oxidation of lipids during exercise as measured with graded calorimetry [117]. 

Fat intake also influences fat oxidation during exercise [3]. A low-fat diet (people randomly assigned to eat 2 or 22% of energy from fat) decreases it by 27% after 2 weeks, parallel to a decrease in the concentration of muscular triglycerides and increased muscle glycogen concentrations compared to the diet containing 22% fat [118]. By contrast, high-fat diets increase lipid oxidation in muscle upon the enhancing effect of a high-fat diet on lipid oxidation during exercise, and this effect takes longer to appear than the almost immediate adaptation to an increase in carbohydrate intake [111]. Alterations in the composition of the fat diet in various lipids do not markedly influence muscle metabolism [184]. Whether a high-fat diet could be a strategy for improving endurance in athletes, via this effect on lipid oxidation, has been largely investigated over the two last decades [185]. In fact, this fat-diet-induced shift in substrate oxidation has been shown to impair endurance exercise metabolism and performance despite enhanced glycogen availability, and thus has little effect on athletes [186]. Furthermore, this aspect has been mostly investigated in athletes, whose metabolism involves important rates of lipid oxidation, and it is clear, on the other hand, that a high-fat diet induces insulin resistance and thus impairs carbohydrate metabolism [187,188,189], whether this fat is saturated or unsaturated [190]. Therefore, the increase in lipid oxidation induced by an increase in fat intake is likely even less interesting in sedentary individuals than in athletes.

The effect of regular protein intake has also been described. A moderate increase in protein up to 1.2 g·kg^−1^·d^−1^ increases MFO [151]. This effect of protein intake may have clinical relevance since it has been reported that the weight-lowering effect of 1 year of exercise training targeted toward lipid oxidation is increased in patients who take egg protein powder in order to reach this value of 1.2 g·kg^−1^·d^−1^ [191].

Finally, prolonged fasting also increases the ability to oxidize lipids during exercise. In a recent study by Frandsen and coworkers [150], MFO increases by 45% after 22 h of fasting and this increase is closely associated with a fasting-induced rise (+114%) in circulating free fatty acid concentrations. This mechanism likely explains what was reported during the Ramadan fast, which shifts lipid oxidation toward higher intensities and increases MFO by 33% [124]. By contrast, in Anorexia Nervosa, which is a situation related to fasting, the LIPOXmax occurs at a lower intensity, although MFO expressed as mg/min/kg muscle mass is the same as in matched controls [152].

### 4.5. Hormones 

Several hormones involved in lipid metabolism have been reported to play a role in the regulation of the balance of substrates during exercise (Figure 6). The kinetics of the appearance of the two catecholamines in the blood are parallel to the shift in the balance of substrates. Norepinephrine rises in the zone of intensities where lipid oxidation is maximal, while epinephrine appears at higher intensities [192]. Norepinephrine mostly activates α-adrenergic receptors while epinephrine equally stimulates α and β-receptors [193]. Via β-receptors, catecholamines stimulate glucose oxidation [194] while α1-adrenergic receptor activation promotes the activation of stimulating mitochondrial energetic molecules (PPARδ, AMPK, and PGC-1α) in various tissues including skeletal muscle [195] and therefore stimulates lipid oxidation [196]. Correlations between the concentration of epinephrine and carbohydrate oxidation during exercise are reported [45], and direct experimentation confirms that epinephrine during exercise increases glucose oxidation [197]. By contrast, the role of norepinephrine in the balance of substrates remains unclear. Indeed, since norepinephrine is an α1-adrenergic receptor agonist and α1-adrenergic receptors stimulate lipid oxidation [195,196], this question probably requires more investigation. Little is known about the effects of cortisol and β-endorphin, which are major stimulators of lipolysis [198,199]. By contrast, the growth hormone (GH), another powerful stimulator of lipolysis [200], stimulates triacylglycerol uptake and storage by the muscle and liver [201] and increases whole-body lipid oxidation and nonoxidative glucose utilization, [202]. GH-deficient individuals have a lower LIPOXmax and MFO that is restored after GH treatment [203]. Downstream GH, insulin-like growth factor I (IGF-I), which enhances lipolysis [204], has also been reported to be correlated with LIPOXmax in soccer players [175]. This correlation may indicate that endurance training simultaneously increases IGF-I release and intramuscular lipid oxidative pathways, or that enhanced function of the somatotrophic axis (GH and IGF-I) during training results in increased muscular lipid oxidation. Interleukin-6 (IL-6) released by the adipose tissue and the muscle is an energy sensor that activates AMP-kinase, and therefore increases glucose disposal, lipolysis, and fat oxidation [205]. The newly discovered hormone adiponectin also increases muscular lipid oxidation via phosphorylation of the AMP-kinase [206]. The adipokine leptin increases muscle fat oxidation and decreases muscle fat uptake, thereby decreasing intramyocellular lipid stores [206].

The effects of sex hormones on fat metabolism may explain why women oxidize slightly more lipids [128,136,181]. During the luteal compared to the follicular phase, there is greater use of lipids and reduced use of CHO, correlated with circulating values of 17-β-estradiol [125,132]. Estradiol and progesterone seem to have opposite effects on the balance of substrates, with greater lipid oxidation when estradiol is used alone [130]. All this leads to the proposition that there is a sexual dimorphism for substrate metabolism during exercise, as recently reviewed by Boisseau and Isacco [131]. Other endocrine axes are certainly also involved in this regulation, but this issue is poorly understood and remains to be studied.

## 5. Targeting Exercise Training on Lipid Oxidation

Initial pilot studies investigating the effects of training targeted at the LIPOXmax in obese patients [7,8,9] evidenced, over a short period of several weeks, some improvement effect on fat mass and other parameters such as the blood lipid profile and blood pressure [35,37], but despite a meta-analysis confirming the efficacy of this approach [207], this did not raise widespread interest. Until now, exercise training targeted toward lipid oxidation did not appear in guidelines for exercise prescription in obesity, diabetes, and other metabolic disturbances.

However, research on exercise in metabolic disorders has, over the two last decades, emphasized the unexpected efficacy of low- to moderate-intensity endurance training [19], showing that despite the low-calorie expense it elicits, it is powerful for controlling weight and improving carbohydrate homeostasis [20,208]. These advances have focused new interest on targeting lipid oxidation. A recent meta-analysis [16] clearly demonstrated that training targeted toward lipid oxidation is efficient for losing weight over a couple of months. Studies of a longer duration are scarce, but a cohort followed by our unit for more than 10 years shows that the weight-reducing effect of this training is long-lasting and sustained over at least 90 months. In Table 2, we present an update of the studies of LIPOXmax training currently published.

### 5.1. Exercise (Regardless of Its Mode or Intensity) Is a Powerful Therapeutic Tool in Metabolic Diseases

Before focusing more precisely on training targeting lipid oxidation, it is important to note that exercise on its own is now unanimously recognized as an efficient tool for preventing the onset of type 2 diabetes [232,233], improving blood glucose control [234,235], and preventing further weight regain in weight-reduced obese individuals [163].

Whether it was a treatment for obesity on its own was not unanimously recognized since diet was considered the cornerstone in the management of weight disorders. However, it has now been convincingly demonstrated that even without any change in diet, exercise may reduce body weight on its own [236,237,238]. 

Regardless of its effects on body weight, exercise is also beneficial for cardiovascular health, due to its beneficial effects on blood pressure, blood lipids, inflammation, blood viscosity, mood, and cognitive function [239]. The exercise-induced metabolic changes that underlie all these effects are evidenced by metabolomic studies and include enhanced oxidative phosphorylation, glycolysis, tricarboxylic acid cycle function, aminoacyl-tRNA biosynthesis, the urea cycle, arginine biosynthesis, branch-chain amino acids, and active skeletal muscle biosynthesis. They are also associated with marked changes in lipid and estrogen metabolism [240].

### 5.2. The Surprising Efficacy of Low-to Moderate Intensity Training

Many national physical activity guidelines indicate that moderate- to vigorous-intensity physical activity should be recommended for cardiovascular health benefits [241]. However, it is clear that the majority of the population does not engage in such physical activity of sufficient intensity and volume [242].

Therefore, it is interesting to consider the (unexpectedly) powerful effects of low-intensity exercise on health. For example, it has been recently reported in the PROOF Cohort Study that French, sedentary, retired individuals over 65 years of age who engage in light physical activity exhibit a 32% reduction in mortality from myocardial infarction and stroke [243]. Rather similar results have been reported in 5314 male US veterans aged 65 to 92 years followed over 20 years, in whom each 1-MET increase in exercise capacity resulted in a decrease in the adjusted hazard of death by 12% [244].

There is now strong evidence that major improvements in insulin sensitivity and glucose disposal can be obtained with light- to moderate-intensity aerobic exercise [20,208]. Exercise induces several changes in the expression of key genes in the skeletal muscle of type 2 diabetic patients. A self-supervised program of walking (>150 min per week) over 4 months improved blood pressure and plasma insulin levels parallel to changes in the muscle proteins that govern mitochondrial biogenesis and metabolism [245].

Low-intensity exercise training also results in improvements in the autonomic nervous system as assessed by the measurement of heart rate variability [246]. Moreover, a program including low-intensity endurance training and coordination improves psychological variables, pain perception, and quality of life in women suffering from fibromyalgia [247].

In pregnant women, moderate-intensity physical activity has been shown to decrease the risk of excessive gestational weight gain, gestational diabetes, and postpartum depression [248]. Effects on the risk of preeclampsia and gestational hypertension are also hypothesized but less well-demonstrated [248]. Low- to moderate-intensity exercise during pregnancy is also a way to improve glycemic control in women with gestational diabetes [249].

Low-intensity exercise is also useful in end-stage chronic kidney disease. In this case, it improves aerobic and functional capacity and therefore improves the quality of life, with no negative impact on renal function [250,251].

In women suffering from menstrual pain, the efficacy of regular exercise (45 to 60 min each time, three times per week or more), even at a low intensity, has been shown to provide a clinically significant reduction in menstrual pain intensity [252].

Similarly, low-intensity exercise improves hypertension. A recent review emphasizes the potential of prescribing patients safe and effective whole-body aerobic exercise at a moderate intensity (i.e., 50–65% of maximum oxygen intake, 30–60 min per session, 3–4 times a week) [253].

Exercise is also beneficial for bone density in postmenopausal women. A recent meta-analysis including 63 interventions at various intensities shows that low- and moderate-intensity exercise are equally effective for the femoral neck, while in this case, high-intensity exercise does not show any efficacy. Moderate-intensity exercise increased total hipbone mineral density (but low-intensity did not), and at the lumbar spine level, all intensities improved bone mineral density but high-intensity exercise yielded greater effects than low-intensity exercise [254].

Another interesting review shows that low-intensity exercise is beneficial for recovering from exercise-induced muscle damage [255].

### 5.3. Training at the LIPOXmax for the Treatment of Obesity

#### 5.3.1. Effects of LIPOXmax Training on Obesity and Associated Disorders in Obese and Overweight Subjects 

LIPOXmax training represents a variety of low- to moderate-intensity exercise training, which is not targeted toward a given level of VO_2max_, but rather toward a metabolic process that occurs at a given intensity. Therefore, as indicated in the introduction, the concept of maximal lipid oxidation was primarily developed in the context of obesity research. It seemed logical to prescribe a variety of exercise that favors the oxidation of lipids in a disease characterized by excess lipid stores. The first results were summarized in a meta-analysis published in 2010 that included 16 studies with training protocols containing as few as two or three sessions per week and a total of 247 participants. The results of this meta-analysis demonstrate the efficiency of training targeted at the LIPOXmax on weight loss, even over a short time period. Body weight decreased by −2.9 kg, together with a decrease in fat mass and waist circumference, as well as an improvement in the ability to oxidize lipids during exercise. Since two-thirds of the studies were performed without an added diet, this meta-analysis demonstrated that LIPOXmax training on its own decreases body fat even if no specific diet is applied [207].

Importantly, ten years later, a novel meta-analysis fully confirms those pilot results. The authors selected 11 trials from 356 publications. Since most of the studies are of short duration, the effect over a period of eight to twenty weeks is analyzed. Over this period, there was a decrease in body weight (−4.30 kg), fat mass (−4.03 kg), and waist circumference (−3.34 cm), while fat-free mass remained unchanged. There was also an increase in maximal oxygen consumption [16].

Of course, those studies performed over a couple of months demonstrated that this variety of training had some effect, but they did not provide any insight into what would occur in the long term.

There are now papers showing that the effect consistently evidenced in the two meta-analyses only represents the early phases of a longer story. Drapier and coworkers’ study [17] presents the effects of 3 × 45 min/week of LIPOXmax training in 88 subjects over 3 years compared to a group on a low-fat diet without exercise and a control group without any exercise or diet. While controls gained weight over these 3 years, both the diet and exercise groups the same weight loss exhibited over the first year. Over the second the third years, however, there was weight regain in the low-fat diet group while the LIPOXmax group continued losing weight (−8.49 ± 2.39 kg at the end of the study). In this study, the measurement of lipid oxidation during exercise before the onset of the study was a predictor of weight loss. Low-fat oxidizers exhibited a lower fat loss. Therefore, Drapier and coworkers’ study showed that this low-intensity exercise training targeted toward lipids maintained its weight-reducing effect for 3 years while diet was no longer efficient, and that this effect was related, in the first year, to the muscular ability to oxidize lipids [17]. 

Thus, although diet procedures are almost always followed by weight regain, the decrease in fat mass reported during the first couple of months was sustained over the long term after 2 years with a parallel loss of trunk and limb fat [18], which was still found after 4 years [255].

The longest duration reported for a follow-up of LIPOXmax training is 96 months [33]. At this time, the patients maintained a weight loss of −9.1 ± 2.3 kg (Figure 7). The reduction of adipose stores equally involves trunk and limb fat and is reflected by a sustained decrease in waist circumference (−5.54 ± 1.96 cm) but not hip circumference. Furthermore, the increase in maximum lipid oxidation capacity was still observed, and there was a slight decrease in diastolic and mean blood pressure, correlated with body composition improvements [33]. 

This finding of a long-lasting efficacy of LIPOXmax training is consistent with one of the conclusions of Baillot and coworkers’ meta-analysis indicating that long-term exercise training interventions result in superior weight loss (11.3 kg) compared to short-term (7.2 kg) and intermediate-term (8.0 kg) interventions [238].

Dandanell and coworkers have recently reported that the ability to maintain a long-term significant weight loss after a lifestyle intervention, which is clearly beneficial for cardiometabolic health [257], is related to MFO [258], as also reported by Drapier and coworkers [17].

#### 5.3.2. Sparing Effects of LIPOXmax Training on Fat-Free Mass (FFM)

In studies conducted over 12 months or less, LIPOXmax training maintains fat-free mass [16,34,35,36,37], and in some publications even increases it [210].

Although the most recognized procedure to improve FFM is to perform resistance training [259,260], there is some literature demonstrating that low-intensity exercise (e.g., a 45 min walk on a treadmill at 40% VO_2_peak) protects lean mass and prevents muscle protein breakdown [261]. In calorie-restrained postmenopausal women, lean mass is equally preserved with both low-intensity (45–50% of heart rate reserve (HRR)) and vigorous-intensity (70–75% of HRR) exercise for a matched total energy expenditure of 700 kcal/wk [262] and similar weight loss. 

Both resistance (RE) and endurance (EE) exercise stimulate skeletal muscle protein synthesis, but the phenotypes induced by RE (myofibrillar protein accretion) and EE (mitochondrial expression) training are not the same, likely due to a differential stimulation of myofibrillar and mitochondrial protein synthesis [263].

This protective effect of low-intensity training may be favored by its glycogen sparing effect, since carbohydrate availability influences muscle protein synthesis and degradation during prolonged exercise [264].

On the whole, although more investigation is required, LIPOXmax training is an efficient way to maintain or even improve fat-free mass by increasing the mass of metabolically active muscle. At the beginning of training protocols in very sedentary patients, it may be used for this purpose. This was the rationale for studies on training in undernutrition situations such as anorexia nervosa, which are still in progress at this time [152].

However, in obese patients engaged in a 2-year training procedure, there was a decrease in fat-free mass [18]. This decrease only involves its non-muscular component (−2.2 ± 0.7 kg), and muscle mass appears to be maintained. 

This long-term exercise-induced decrease in FFM needs to be interpreted in light of the findings of the team of Isabelle Dionne who showed that a greater fat-free mass in sedentary patients is associated with insulin resistance and not with better insulin sensitivity [265,266]. Such a decrease in fat-free mass after long-term training, which improves carbohydrate homeostasis, low-grade inflammation, and blood pressure, should probably not be considered a deleterious effect. The dramatic increase in the ability to oxidize fat expressed by kg of muscle mass after 96 months [33] supports this assumption. 

#### 5.3.3. Metabolic Effects of LIPOXmax Training

MFO and LIPOXmax have been found to be correlated with insulin sensitivity in middle-aged male obese subjects, both of them correlated with mitochondrial pyruvate oxidation and mitochondrial density [267,268]. Multivariate analysis indicates that insulin sensitivity is statistically better ‘explained’ by the maximal ability to oxidize lipids, which is, in turn, better explained by the mitochondrial function parameter Vmax/V0 pyruvate [267,268].

Thus, insulin sensitivity, mitochondrial function, and lipid oxidation during exercise are closely related. Furthermore, a study on healthy young men shows that insulin sensitivity is positively correlated with lipid oxidation during exercise, which is, in turn, correlated with 24 h fat oxidation. Therefore, a high capacity to oxidize fat is likely to be advantageous for metabolic health [269].

There is also an inverse association between maximal lipid oxidation during exercise and the fatty liver index [141], which is known to be associated with insulin resistance that, in turn, induces an increase in free fatty acids responsible for mitochondrial dysfunction [270]. This is, of course, another component of the constellation of disorders that link insulin resistance to decreased oxidation of fat at rest and during exercise.

It is logical to assume that training targeted toward the LIPOXmax decreases insulin resistance, and this was reported in a study of 2-month training in patients with metabolic syndrome [209].

In fact, in the initial ten weeks of training, the first improvement observed parallel to the increase in MFO was a reversal of the decrease in first-phase insulin secretion [256]. This means that exercise training targeted toward maximal lipid oxidation improves pancreatic beta-cell function earlier than insulin sensitivity, which requires a significant reduction in fat mass before it is corrected. Other studies also show that exercise training rapidly restores the early-phase insulin secretion, which is usually blunted in adults with prediabetes [271], an effect that can be obtained after 12 weeks of low-intensity training targeted toward the lactate threshold, i.e., slightly above the LIPOXmax [272]. This favorable effect of exercise training on beta cell function better predicts the improvement in glucose homeostasis than insulin sensitivity [273]. Recent research helps to understand this finding, since it has shown that human skeletal muscle cells secrete myokines such as CX3CL1 (fractalkine), which protect β-cells from the negative impact of TNFα [274], and that short-term training improves β-cell function and efficiently reduces ectopic fat within the pancreas in prediabetic or T2D patients [275]. In addition, studies performed on T2D suggest that long-duration higher-intensity exercise is less beneficial than moderate-duration and -intensity exercises [276,277].

#### 5.3.4. LIPOXmax Training Improves Inflammatory Status

Low-grade systemic inflammation is suggested to play a role in the development of a host of chronic diseases including obesity, diabetes, and cancer. A number of studies suggest that in these diseases, regular exercise has anti-inflammatory effects so physical activity as such may suppress systemic low-grade inflammation [278,279,280]. On the whole, both endurance and resistance training are known to decrease C-Reactive Protein (CRP) [281]. In two studies, LIPOXmax training has been shown to decrease CRP [282,283]. In these studies, changes in CRP were negatively related to those of lipid oxidation during exercise suggesting that the improvement in the ability to oxidize lipids during exercise is associated with an anti-inflammatory effect. Further studies are needed. A similar effect was found in LIPOXmax reunion [222] but this part of the study is still unpublished. 

In a recent study by Gram and coworkers [284], an anti-inflammatory effect of exercise training is only evidenced for active commuting and moderate-intensity leisure time exercise (~70% VO_2peak_), but not vigorous-intensity leisure time exercise (~70% VO_2peak_). 

Although more investigation is needed regarding this issue, convincing evidence suggests that low-intensity regular exercise, and of course LIPOXmax training, are able to reverse low-grade inflammation. 

#### 5.3.5. LIPOXmax Training Also Decreases Blood Viscosity

Regular practice of exercise decreases blood viscosity factors and increases blood fluidity [285]. This effect has been evidenced by training at a low intensity targeted toward lipid oxidation [7].

More precisely, the hemorheological parameter, which is significantly decreased by LIPOXmax training, is plasma viscosity. There is a continuum for plasma viscosity, which is lower in athletes than in low-intensity-trained, and even more sedentary, individuals. As humans since prehistoric periods performed a great deal of daily low-intensity exercise at this level, it is logical to assume that it was a part of the “healthy primitive lifestyle”, and thus, that a decrease in plasma viscosity is one of the biological consequences of this lifestyle, likely with some relevance to cardiovascular health [85].

### 5.4. LIPOXmax Training in Adolescents Suffering from Obesity

While puberty downregulates MFO and LIPOXmax [120], obese teenagers exhibit a lower MFO [46,183]. However, their LIPOXmax occurs at a similar intensity to that of nonobese matched teenagers (41% VO_2max_, i.e., 58% HRmax) [286].

After two pilot studies of LIPOXmax training in obese teenagers by Brandou and coworkers [8,210], Lazzer and coworkers compared the effects of three different exercise training protocols over 3 weeks in 30 obese adolescents. The study shows that low-intensity training (40% VO_2max_) induced significantly greater weight loss than high-intensity (HI, 70% VO_2max_) and high-intensity interval training (HIIT): Crude weight: −8.4 ± 1.5 vs. −6.3 ± 1.9 vs. −4.9 ± 1.3 kg and fat mass: −4.2 ± 1.9 vs. −2.8 ± 1.2 vs. −2.3 ± 1.4 kg, respectively, while HI and HIIT induced a greater increase in VO_2_peak and the fat oxidation rate [287]. A series of studies by Tabka and coworkers [212,215,216,217,219] confirmed the fair efficacy of daily LIPOXmax training over 2 months associated with a hypocaloric diet in Tunisian obese adolescents.

### 5.5. Bariatric Surgery

Bariatric surgery has, over the last decade, been the major advance in the management of severe obesity. After losing weight with this technique, MFO increases, but the power at which lipid oxidation reaches its (LIPOXmax) is shifted toward lower intensities [138,288].

Therefore, the question arose of whether taking into account this left shift of the LIPOXmax for targeting exercise would have an effect on these patients after surgery, in the long term. In the first study, conducted over 40 months after sleeve gastrectomy, we evidenced an additive effect that appears after 1 year. Patients exercising at the level of the LIPOXmax (3 × 45 min per week) continued gradually losing weight, while stabilization was observed in the untrained group [289]. At 60 months post-sleeve, the difference between the two cohorts became even more impressive since weight loss continued in the trained group while a tendency of weight regains was observed in the untrained group. At 60 months post-sleeve, exercising patients lost, on average, 53 kg while non-exercising (control) post-sleeve patients were only 25 kg below their initial weight [290].

Synergy between bariatric surgery and regular exercise is now well-recognized [291], but most studies are of a short duration. Our cohort shows that low-intensity exercise targeted toward lipid oxidation, which is well-tolerated and easy to prescribe and follow in these patients, is a powerful add-on to sleeve gastrectomy in individuals who, before surgery, reported no diet or exercise strategy could control their weight. The reason for this efficacy is not clear. One week after the onset of LIPOXmax training sessions post-sleeve, a decrease in orexigenic pulsions and an increase in satiety scores can be evidenced [290].

### 5.6. Diabetes

Although there is also a report of alterations of the balance of substrates in type 1 diabetics who exhibit a lower ability to oxidize fat during exercise [147], we mainly focus on type 2 diabetes since, in this disease, physical activity is considered treatment as efficient as drugs [292]. Both endurance and resistance training can improve blood glucose control, achieving a reduction in HbA_1c_ by 0.6 to 0.8% and thus should be prescribed as a true treatment [293].

Several studies show that exercise training targeted toward the LIPOXmax is able to decrease HbA_1c_ [220,222]. Follow-up over 3 years of targeted physical activity at the LIPOXmax (3 × 45 min per week) in 16 type 2 diabetics compared to 287 nondiabetic matched controls resulted in a decrease in HbA_1c_, which remained at a mean of −0.65 ± 0.34% after 3 years, but there was a weight regain in diabetics in contrast with the continuation of weight loss in non-diabetics. Diabetic patients not treated with sulfonylureas and/or insulin lost more weight than those receiving these treatments. Therefore, LIPOXmax training, which is able to reduce weight in the long term in nondiabetics, has not sustained this weight-lowering effect over 3 years in type 2 diabetics, even if patients are treated with insulin and/or sulfonylureas. However, it appears to improve blood glucose control [294].

An attractive concept derived from the physiology of the balance of substrates was proposed in 2015 by Francescato and coworkers [295], termed «glucose pulse». Given the almost linear relationship between exercise intensity and carbohydrate oxidation by the muscle, it was assumed that the heart rate during exercise can be used as a predictor of glucose disposal and, thus, blood glucose control. In fact, in diabetics performing a steady-state exercise bout that gradually decreased glycemia, it was found that CHO oxidation rates measured with calorimetry did not predict the decline of blood glucose values at all, challenging this attractive concept [296].

### 5.7. LIPOXmax Training in Patients Gaining Weight under Psychotropic Drugs

Another situation that induces overweight is heavy psychotropic treatment with neuroleptics in mental illness. Most of those drugs induce weight gain, and there is no obvious strategy to avoid this. A 3-month training program at the LIPOXmax was able to prevent weight gain in patients who seriously engaged in training, while others who did not adhere to the exercise protocol continued to gain weight. Therefore, low-intensity endurance training targeted at the level of the LIPOXmax appears to be effective in the prevention of psychotropic-drug-induced weight gain. Further studies are needed to confirm this finding [218].

### 5.8. LIPOXmax Training in Cancer

After treatment for breast cancer, the balance of substrates oxidized during exercise is disturbed, with a lower ability to oxidize lipids [297]. It is known that exercise is important after treatment for cancer to improve life expectancy and the quality of life. Low-intensity exercise is one of the strategies proposed, although the literature rather supports supervised, moderate- to high-intensity, combined resistance, and aerobic exercise training. Low-intensity physical activity that can be performed at home is considered a viable alternative for women who cannot follow a higher-intensity protocol. In this respect, there is a rationale to use LIPOXmax training due to its metabolic and anti-inflammatory effects, which are also well-tolerated in such asthenic patients [298].

### 5.9. Osteoarthritis

There is no report of exercise specifically targeted at the LIPOXmax in osteoarthritis, but it is striking to note that sedentary people who complain of joint pain at the beginning of LIPOXmax training usually undergo rapid improvement, which helps them to adhere to a long-term training program. The literature on exercise in osteoarthritis [299,300] indicates that low-intensity aerobic exercise (walking or cycling) is one of the most effective strategies for this disease. Further studies, more precisely taking into account the balance of substrates, may perhaps be interesting in this context.

### 5.10. How to Follow This Low-Intensity Training?

It is now recognized that obesity itself can be treated as a chronic condition, with follow-up support contributing to the maintenance of weight loss [301]. Various help such as connected devices can be used, but clearly, regular outpatient consultations are the simplest and most efficient procedure to date to obtain good results in the long term [302]. In a recent study, Bughin and coworkers [231] found in 49 LIPOXmax-trained patients, randomized to conventional follow-up vs. a mobile telerehabilitation solution on a smartphone, that telerehabilitation was not superior to the usual care for improving body composition. 

Another issue is the frequency of exercise sessions. In the KAROLA study, which presents the follow-up of a training program in patients with stable coronary heart disease over 10 years, the best adherence in the long term was obtained with 3–4 sessions per week, while higher frequencies were not maintained over the long term [303].

Therefore, regular follow-up together with a realistic training volume is a key factor for obtaining long-term adherence to training. 

Obviously, nutrition also matters. Many studies add a restrictive diet to exercise, but there are more and more concerns emerging about restrictive diets, which, in the long term, induce resistance to weight loss due to homeostatic adaptations [304]. Rather than restriction, careful equilibration of the diet can be proposed. In this respect, maintaining an adequate protein intake is likely an important point [305]. During a restricted diet in senior people with metabolic syndrome, if exercise training is also performed, a lower threshold intake for protein must be set at 1.2 g·kg^−1^·d^−1^ to maintain blood protein homeostasis [306]. According to Soenen and coworkers, a daily protein intake of 1.2 g·kg^−1^·d^−1^ is necessary for the preservation of resting energy expenditure and FFM, as well as for lowering blood pressure [307]. Increasing protein intake at this level of 1.2 g·kg^−1^·d^−1^ improves MFO [151]. Moderate protein enrichment of the diet targeted at 1.2 g·kg^−1^·d^−1^ in obese women exercising at the LIPOXmax has a synergistic effect with exercise training, inducing greater weight loss compared to training at LIPOXmax only or protein enrichment only [191]. We thus propose, during LIPOXmax training, a dietary check-up and counseling to avoid restriction, with special emphasis on the need to maintain an adequate protein intake, rather than a restricted diet.

## 6. How Can We Explain the Unexpected Efficacy of Low Volumes of Low-Intensity Exercise?

The very low energetic expense elicited by LIPOXmax training likely explains why most scientists do not believe that it can have any efficacy. We have seen that this kind of training improves mitochondrial function and eating behavior, and the recent literature on low-intensity exercise shows that it elicits other specific effects.

As an introduction to this chapter, we should note two important points. First, aerobic and resistance training do not stimulate the same signaling pathways and therefore induce different myofiber phenotypes. Grossly, aerobic training stimulates the activation of slow-fiber gene expression including genes encoding proteins involved in mitochondrial biogenesis and energy metabolism, while resistance training rather promotes the expression of genes coding for contractile proteins [308]. On the other hand, the response to various exercise protocols differs between metabolically impaired and healthy individuals [309].

In addition, regarding aerobic exercise training, studies that compare low-intensity and high-intensity exercises indicate that they are different tools with entirely different biological effects.

This is clearly evidenced in a study comparing interval training and LIPOXmax training in 63 type-2 diabetics over a period of 3 months, without nutritional intervention [220]. After 3 months, both training procedures increased VO_2max_ but interval training was much more efficient for this (+42% vs. +14% for LIPOXmax training). Interval training reduced resting systolic blood pressure (−12 mmHg) and total cholesterol (−0.29 mmol/L), while LIPOXmax training did not. Both procedures decreased overweight, but only the LIPOXmax training improved MFO and shifted the LIPOXmax to a higher power intensity. LIPOXmax training decreased fat mass (−1 kg), increased fat-free mass (+1 kg), and decreased waist circumference (−3.8 cm) and hip circumference (−2.2 cm), while interval training did not significantly change any of those parameters, and the effects of training on HbA_1c_ were significant in the LIPOXmax group (−0.15%) but not in the interval training group. The interest of this study is to clearly show that one of these procedures is not better than the other, but that they simply have different effects. Interval training is very efficient for improving aerobic working capacity, blood pressure, and the lipid profile, while low-intensity endurance training (LIPOXmax training) improves the ability to oxidize lipids during exercise, increases fat-free mass, decreases fat mass, and decreases HbA_1c_. The benefits of these two procedures are, thus, quite different.

### 6.1. LIPOXmax Training Improves Mitochondrial Respiration and the Krebs Cycle Function

As already explained above, exercise training targeted toward the LIPOXmax exhibits marked effects on whole-body lipid oxidation and mitochondrial respiration, both of which were correlated [86,89]. 

This effect of LIPOXmax training on the mitochondria is the first explanation of its efficacy despite its low intensity and low impact on energy deficit. 

### 6.2. Secretion of Chemical Messengers (‘Myometabokines’) by Muscle

More recently, in muscle tissue from participants in the STRRIDE studies, the metabolite signatures of exercise training have been investigated [22], showing increased concentrations of the tricarboxylic acid cycle intermediates. Among them, succinate and succinylcarnitine emerged as the strongest correlate of insulin sensitivity, which, in this study, is improved by low-volume, low-intensity exercise. Succinate and succinylcarnitine are tricarboxylic acid cycle intermediates, which can act as chemical messengers in the body and are termed myometabokines [26]. This means that, regardless of other mechanisms, the low-volume low-intensity protocol was efficient enough to generate, through the activation of the tricarboxylic cycle, active metabolites, which are likely to mediate an improvement in insulin sensitivity. 

In this respect, it is important to point out that there are rather differential effects of moderate- and vigorous-intensity exercise on key muscle proteins in the glucose metabolic pathway [310].

### 6.3. Sedentarity and Exercise as Epigenetic Programmers of Various Functions in the Body

There is now a great deal of information demonstrating that a lack of physical activity and long periods of sitting induce insulin resistance and increase blood pressure, shifting the lipid profile toward an atherogenetic one and thus increasing the risk for atherothombosis and increased mortality [311,312,313,314]. These studies point out the importance of breaks in prolonged sitting periods, in order to prevent these health concerns. Interestingly, the replacement of modest amounts of sitting time with even light physical activity has the potential to reduce the risk of premature death among less-active adults [315].

Until recently, the effects of sedentarity and exercise on body composition and biological functions were assumed to be explained by the energetic balance [12,316]. However, it has since become obvious that exercise is more efficient for controlling body weight than would be expected from the energy deficit alone [31,236,317].

Metabolic defects underlying insulin resistance and impaired fatty acid oxidation in skeletal muscle remain intact in muscle cells raised in culture [318], leading to the hypothesis that the skeletal muscle of severely obese sedentary individuals displays a constitutive “obesity metabolic program”, which contributes to the positive lipid balance (weight gain) and insulin resistance. This program is not completely corrected by massive weight loss induced by bariatric surgery but is completely corrected by exercise training [319]. Therefore, the development of severe obesity in response to a sedentary lifestyle appears to be driven by a series of changes in gene expression that can be accurately described as a program that leads to fat storage and is reversed by physical activity [320,321].

These authors assumed that this “program” consisted of epigenetic alterations, i.e., age- and lifestyle-dependent changes in gene expression that occur without nucleotide alterations in the DNA coding sequence [322].

Epigenetic alterations that underlie the “obesity program” have been investigated, leading to a rather complicated picture. Several epigenome-wide methylation studies reported an association between BMI and altered methylation sites in genes such as CPT1A, ABCG1, PGC1α, HIF3A, and SREBF1 [323,324,325]. These modifications include histone modifications and DNA methylation, as well as other molecular events, such as short noncoding RNAs (miRNAs), which bind to mRNA to inhibit or impair their translation [326,327].

The direct exposure of individuals to excess nutrition or obesogenic products induces epigenetic changes that are likely to play a role in the development of obesity [328]. Ten days of bed rest also increased DNA methylation of the gene coding for peroxisome proliferator-activated receptor gamma, coactivator 1 alpha (PPARGC1A), a coactivator for the PPAR pathway that governs mitochondrial oxidative metabolism and fiber-type switching in skeletal muscle [329].

Furthermore, exercise exerts short-term transient effects on epigenetic targets [330,331] and seems to have a wider epigenetic impact in the long term during exercise training [332,333,334]. Physical activity (PA) elicits an inflammatory reaction that triggers epigenetic changes [335] that include DNA and histone modifications, as well as the expression of specific microRNAs, and thus can induce specific and fine-tuned changes to the transcriptional response [336].

Nutrition and physical activity represent a powerful potential epigenetic intervention point for effective cardiovascular protective and management strategies [337]. In first-degree healthy relatives of T2D patients, a 6-month endurance exercise intervention decreased DNA methylation in a number of genes with known metabolic function, increasing gene expression in pathways associated with mitochondrial function [338].

### 6.4. Sedentarity and Exercise Effects on Appetite

One of the likely explanations for the paradoxical weight-lowering effect of exercise training at a low intensity is likely its effect on eating behavior. 

The effects of exercise training on eating behavior constitute a very complex issue, which has been extensively studied by teams such as the group of J Blundell. Very interestingly, these authors have shown that exercise exerts a dual effect on appetite control. It can increase the orexigenic drive and improve meal-induced satiety [339]. Since the effects of all exercise sessions appear to be a mix of these two effects, it remains difficult to predict whether a variety of exercises will increase or decrease food intake. This team shows that during a 12-week training protocol, some individuals lost up to 14 kg, while others gained weight. Those who lost weight spontaneously decreased their food intake and those who gained weight increased their food intake [340]. This means that exercise may, according to its effect on appetite and satiety, result in either weight gain or weight loss. 

Across the entire spectrum of exercise intensities, there is a relationship between exercise habits and the amount of ingested food. An up-to date systematic review and meta-analysis [341] shows the accuracy of the old findings of Mayer and coworkers [342] who described, in 1956, a “J-shaped curve” with higher energy intake in sedentary individuals, slightly lower intake when people practice low levels of exercise, and again, higher calorie intake in people exercising heavily.

Despite the paucity of studies on the specific effect of exercise targeted toward lipid oxidation, there is some evidence that it is associated with lower energy intake than both sedentarity and high-intensity endurance exercise [30,343,344].

By contrast, there is a paradoxical effect of low-volume high-intensity exercise, which may increase appetite and thus result in higher food intake and paradoxical weight gain [345] (Figure 8).

The moderating effect of low-intensity exercise training on food intake is also found in individuals trained at the LIPOXmax after sleeve gastrectomy, but is less pronounced than in non-gastrectomized subjects [290] and may play a role in the efficacy of this kind of training for improving weight loss after this surgery.

The mechanisms of these effects on eating behavior are not clear. It is likely that hormones involved in the regulation of food intake play a role, but an effect on glucagon-like peptide 1, peptide YY, and cholecystokinin [346] and the suppression of the orexigenic hormone ghrelin [347] are only found with higher-intensity exercise.

Myokines released during exercise, such as the tricarboxylic acid cycle intermediates (Myometabokines) and, more precisely, succinate [26], may modulate eating behavior among many functions [348,349]. Presumably, exercise using the most fat as possible as a source of energy is able to involve the Krebs cycle to a greater extent than short bouts of high-intensity exercise [350] and thus generate important quantities of these metabolites that exert pleiotropic effects. 

## 7. Other Strategies Based on the Balance of Substrates during Exercise

### 7.1. Targeting Exercise at the Level of the “Point of Crossover”

Recently, coming back to the initial ‘crossover concept’ [43], Coquart and coworkers [351] investigated the concept of targeting slightly higher than LIPOXmax, at the “point of crossover”, i.e., the intensity at which 70% of energy comes from CHO and 30% from fat, as initially proposed by Perez-Martin and coworkers [1]. In postmenopausal obese women exhibiting metabolic syndrome, a 12-week training program at this level (three 45-min sessions/wk on a cycle ergometer) resulted in a decrease in body weight (−2.7 kg), fat mass (−1.4 kg), and waist circumference (−5 cm), together with an improvement in fasting plasma glucose and systolic blood pressure. Values of the surrogate of insulin resistance “Quantitative Insulin Sensitivity Check Index” (QUICKI) and of the score of metabolic syndrome were also improved. Therefore, training at this level is beneficial for body composition, blood pressure, and glucoregulation. Whether its effects are different from those of training targeted toward the LIPOXmax or the ventilatory threshold, which are close to this level (see Section 3.4 above), remains to be established. 

### 7.2. Optimal Level of Fat/Carbohydrate Oxidation Ratio (OLORFOX)

Another recently proposed strategy based on the balance of substrates is to prescribe endurance training, minimizing carbohydrate oxidation by targeting the optimal level of the fat/carbohydrate oxidation ratio (OLORFOX) [352]. This level is calculated from the ratio between fat oxidation and CHO oxidation (Figure 9). In our database of 5258 calorimetries [153], we found that this ratio peaks, on average, 12% below the LIPOXmax but with a great deal of inter-individual variation. In fact, there are profiles of subjects in whom this level is strikingly dissociated from that of LIPOXmax. An example of this situation is hypothyroid patients treated with levothyroxine who oxidize more lipids at mild to moderate intensities and more CHO at high intensities compared to controls [353]. In this population, 3 years of exercise training targeted toward the OLORFOX helped individuals to lose 11.5 ± 1.5% of their initial weight. We, therefore, propose more thorough evaluation of this new concept of the metabolic targeting level of endurance training in individuals exhibiting this biphasic profile [352].

## 8. Concluding Remarks

This review puts together several lines of evidence supporting the concept that the measurement of the balance of substrates during exercise is reproducible and provides information about metabolic disturbances associated with sedentarity and obesity, as well as about adaptations of fuel metabolism in trained athletes. It also becomes obvious that low- to moderate-intensity endurance exercise training, among other exercise protocols, exerts powerful effects, and that when this exercise is more closely targeted toward lipid oxidation, it is useful for weight reduction in patients suffering from obesity and improves blood pressure and the metabolism of carbohydrates and lipids (Figure 10). LIPOXmax training is thus safe, easy to accept and implement over the long term, and has been routinely used by several teams over the world for more than 20 years. Although a great number of pilot studies are now available, many questions remain to be addressed. The goal of this review was thus to gather a large body of information about LIPOXmax training, in order to provide the basis for further research in this area, which is likely to be promising in the fields of nutrition and metabolism as well as sports medicine. Many issues remain to be investigated in order to gain a comprehensive picture of physiological and nutritional factors that modify the balance of substrates. The hormonal influences summarized in Figure 6 remain to be more thoroughly investigated; whether α-1 adrenoreceptor stimulation (possibly triggered by norepinephrine) is or is not an important mechanism behind the increase in lipid oxidation during exercise, as suggested by recent works remains unclear. More training studies are also needed to confirm, and more precisely analyze, the long-term effects of LIPOXmax training on body composition, metabolism, and health, and to determine the effects over periods longer than 8 years. The effects on eating behavior, despite their potential importance as an explanation of the efficacy of LIPOXmax training, have received little attention up to now and are only reported by one team [30,290]. The effects of LIPOXmax training in specific situations such as osteoarthritis, patients treated for cancer, and patients taking psychotropic drugs (see Section 5 of this paper) also remain poorly investigated. The beneficial effects of LIPOXmax training on low-grade inflammation have been reported but need to be more precisely described and explained. The efficacy of LIPOXmax training as an add-on to bariatric surgery and the mechanism that underlies it remain to be more precisely studied [289,290]. Moreover, the myometabokines that actually explain the beneficial effects of LIPOXmax training remain to be precisely determined.

## Figures and Tables

**Figure 1 nutrients-14-01605-f001:**
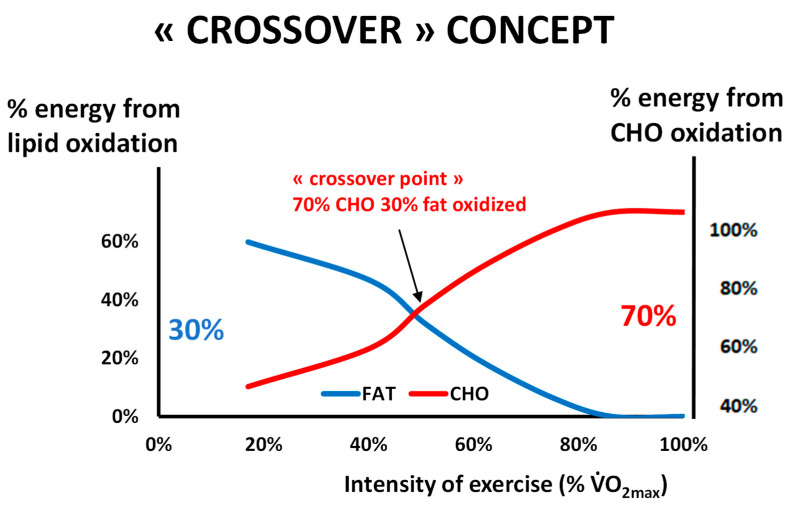
The classical picture of Brooks and Mercier’s “Crossover concept” redrawn from our personal database of more than 5000 exercise calorimetries (see text). This hypothesis assumes that there is a major shift in the balance of substrates used for oxidation during exercise grossly around 50% of the maximal aerobic capacity, when carbohydrates represent more than 70% of the sources of energy for the exercising body. Oxidative use of fat culminates below this level, and close to it or slightly above it, blood lactate increases and the ventilatory threshold occurs. Note that the ordinates for % of fat oxidation and % of CHO oxidation are not symmetric, in order to better visualize the crossover. In a series of more than 5000 exercise calorimetries, we find that the crossover point is, on average, at 55.1% of VO_2max_ but exhibits a wide variability among individuals.

**Figure 2 nutrients-14-01605-f002:**
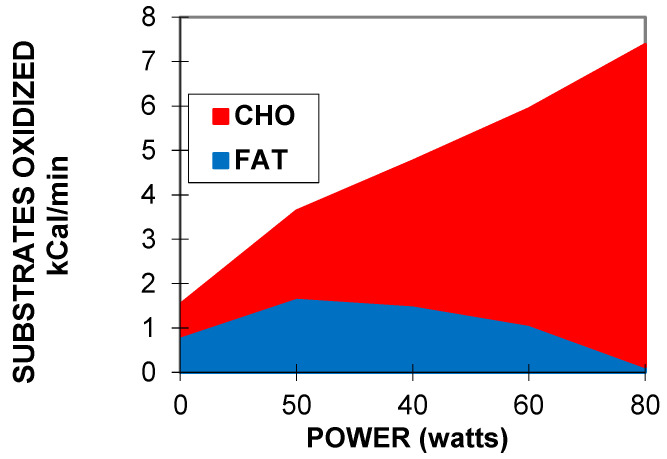
An example of exercise calorimetry performed on a 75-year-old male subject explored for training prescription in grade-2 obesity (weight 99.3 kg, body mass index: 36 kg/m², percentage of fat: 27.7%, VO_2max_ ACSM: 19.7 mL/min/kg). Four 6-min steps were performed: 20, 40, 70, and 100 watts. The crossover point is found at 68 Watts (i.e., 56% of maximal aerobic power) and the LIPOXmax is found at 51 Watts (i.e., 42% of maximal aerobic power). The MFO is 297 mg/min, i.e., 8.2 mg/min/kg of muscle mass. CHO represents 100% of the oxidized substrates above 100 watts, i.e., 82% of maximal aerobic power. It can be seen that over the range of intensities applied during the test, CHO oxidation can be approximately modeled as a straight line (*carbohydrate cost of the watt*) whose slope is, in this individual, 0.24 mg/min/kg/watt.

**Figure 3 nutrients-14-01605-f003:**
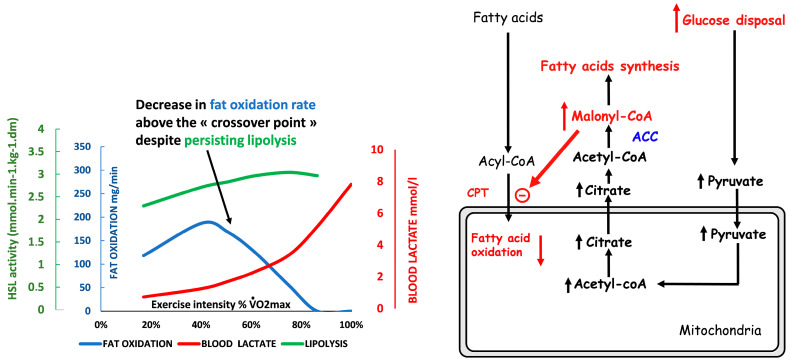
Schematic representation of the events explaining the “bell-shaped curve” of lipid oxidation when exercise intensity increases (see text). Lipolysis is not the limiting step and is still active far above the “crossover point”, but lipid oxidation is inhibited by the metabolites generated by the increase in the rate of CHO oxidation, which becomes the dominant fuel. At this intensity level, blood lactate increases because of the high rate of carbohydrate processing. Therefore, the LIPOXmax (intensity where lipid oxidation reaches its top) occurs below the “crossover point”, the rise in blood lactate, and the ventilatory threshold. The drawing of lipid oxidation is obtained from our database including 5258 calorimetries, and the point of maximal lipid oxidation occurred at 47 ± 1% of VO_2max_. At the level of LIPOXmax, individuals oxidized 209.5 ± 1.37 mg/min of lipids. This level is widely variable among individuals. Lactate data are from [59] and intramuscular lipolysis from [60]. HSL: Hormone-sensitive lipase activity.

**Figure 4 nutrients-14-01605-f004:**
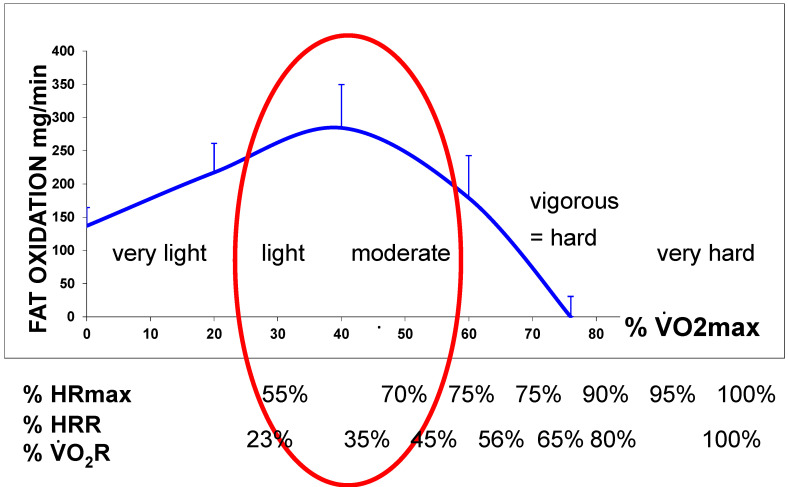
The lipid oxidation zone or “LIPOX zone” corresponds to a level of light to moderate exercise (usually 30–50% of VO_2max_, quoted from “very light” to “moderate” on scales of perceived exertion, and corresponding in practice to 50–70% of the maximal heart rate (HRmax). Equivalences between % of VO_2max_ and % heart rate reserve (HRR) and % VO_2_ reserve are calculated according to [154,155].

**Figure 5 nutrients-14-01605-f005:**
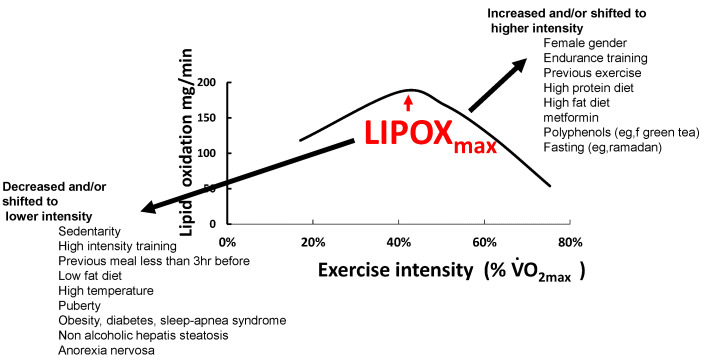
Schematic representation of physiological situations that modify the asymmetrical dome-shaped curve of lipid oxidation.

**Figure 6 nutrients-14-01605-f006:**
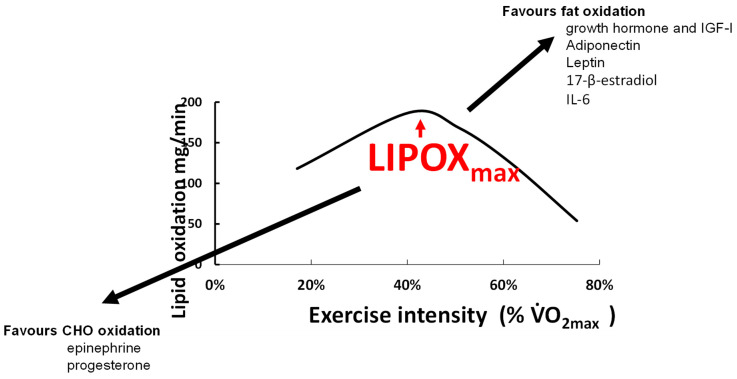
An attempt to summarize hormonal influences that shift the balance of substrates towards more carbohydrate vs. more fat oxidation. As indicated in the text, this issue remains poorly investigated.

**Figure 7 nutrients-14-01605-f007:**
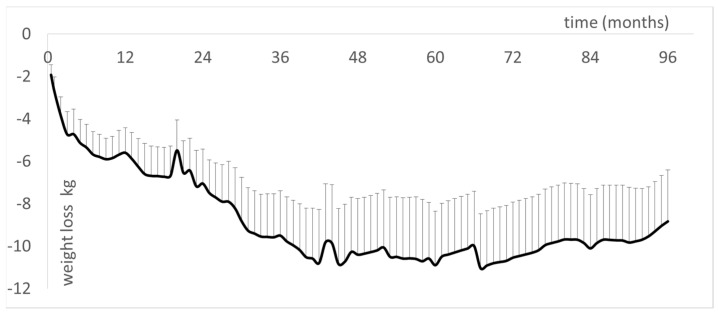
Follow-up of a cohort of 12 obese patients regularly followed over 96 months (results partially presented in [33] and further updated in January 2022). This is the continuation of previously presented shorter follow-up studies [17,18,231,256]. Results show that the weight-lowering effect of LIPOXmax training persists over more than 8 years.

**Figure 8 nutrients-14-01605-f008:**
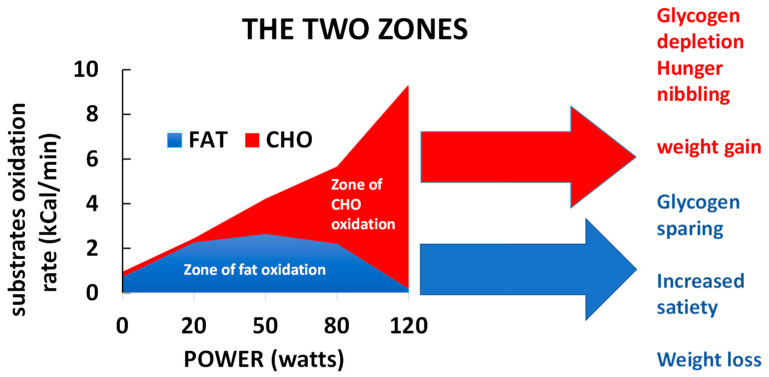
An example of exercise calorimetry in a patient exhibiting a markedly ‘biphasic’ profile of substrate oxidation during exercise. Opposite effects on eating behavior of the two zones of the balance of substrates may, in part, explain why exercise targeted toward fat oxidation induces weight loss [30,344]), while short bouts of exercise targeted toward higher intensities in the zone of predominant oxidation of CHO may induce overeating and paradoxical weight gain [345].

**Figure 9 nutrients-14-01605-f009:**
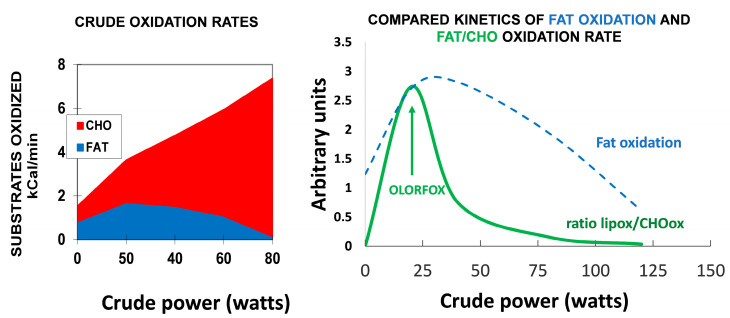
An example of the biphasic profile of substrate oxidation in a 48-year-old woman who suddenly gained weight after discontinuing sport practice. Current weight is 96 kg (percentage of body fat 37.6%). The LIPOXmax occurs at 64 watts (i.e., 42% VO_2max_), and MFO at this level is 282 mg/min. However, the optimal level of fat/carbohydrate oxidation ratio (OLORFOX) [352] occurs at a lower power intensity at 23 watts, so the target heart rate can be proposed as 82 b/min, rather than 92 bpm, if one wants to spare carbohydrates.

**Figure 10 nutrients-14-01605-f010:**
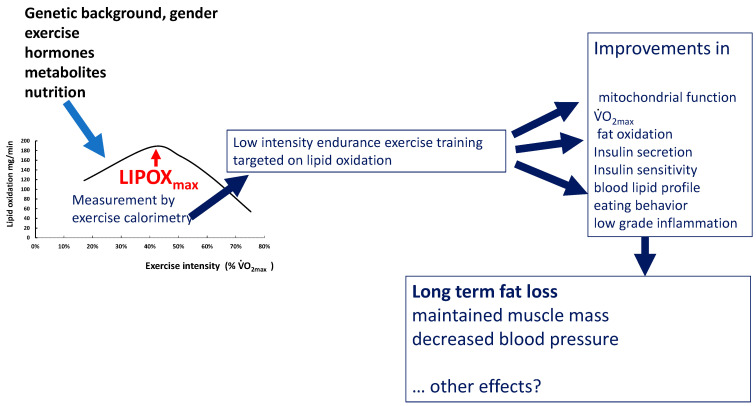
Summary picture. Maximal oxidation during exercise is determined by genetic background and gender, as well as hormones (catecholamines, GH, estradiol), exercise habits, and nutrition. It is a rather stable and reproducible parameter, unless subjects change their lifestyle, and is impaired in various pathologic conditions (obesity, diabetes, sleep apnea, bariatric surgery). This level or others closely related can be used for targeting exercise training. There is now good evidence that in the middle term (2–6 months), this variety of exercise training improves mitochondrial function, VO_2max_, lipid oxidation during exercise, parameters of glucose disposal, LDL cholesterol, and low-grade inflammation. Fat mass decreases and fat-free mass are maintained. In diabetics, the decrease in HbA_1c_ elicited by this variety of training is of the same magnitude as that of other protocols (−0.6%). In fact, the effects on VO_2max_, blood pressure, and circulating levels of lipids at 2–6 months are less pronounced than those of higher-intensity protocols and interval training. A moderating effect on sedentarity-associated overeating is also reported by several studies and occurs over the first week. Long-term effects are only reported by a limited number of nonrandomized studies but they appear to be prolonged (up to 7 years), inducing a gradual improvement in body composition (loss of central and limb fat mass, maintenance of muscle mass) and blood pressure. Whether such prolonged training can have other beneficial effects is not yet known. More studies are needed to better describe and understand all these processes.

**Table 1 nutrients-14-01605-t001:** Factors, diseases, and physiological situations that have been studied for their effect, or lack of effect, on the ability to oxidize lipids during exercise.

Modifying Factor	Effect	References
Previous meal taken less than 3 h before	Decreased MFO and shifted LIPOXmax to a slightly lower intensity	[5,6,111,112,113]
Dietary carbohydrate and fat intake	Dietary carbohydrate and fat intake make modest but independent contributions to the interindividual variability in the capacity to oxidize fat during exercise.	[114]
Polyphenols of green tea	They increase maximal fat oxidation and shifts the point where fat is no longer oxidized to higher intensity levels	[115,116]
Anthocyanins from *Prunus cerasus* L.	Reported to be capable of augmenting fat oxidation but do not modify MFO	[117]
Low-fat diet	Decreases fat oxidation during exercise	[118]
Previous exercise performed just before the exercise calorimetry	1 h single bout of moderate-intensity exercise slightly increases MFO	[119]
Puberty	LIPOXmax and MFO are higher in prepubertal children and gradually decrease throughout puberty to reach adult values at the end of puberty	[120,121,122,123]
Type of exercise	Higher during running than cycling in adults and in pre- to early pubertal children	[63,122]
Ramadan	At the end of Ramadan, subjects had increased their fat utilization during exercise, with a right-shift of the cross-over point and the LIPOXmax and an increase in MFO	[124]
Gender	Women oxidize slightly more lipids, and on average, their LIPOXmax occurs at higher relative output. Estradiol and progesterone seem to have opposite effects, with estradiol eliciting greater lipid oxidation. Greater use of fat and reduced amount of CHO usage during the luteal vs. follicular phase, directly related to the change in estradiol.	[122,125,126,127,128,129,130,131,132]
Temperature	Shift to preferential CHO oxidation during exercise in hot environments. Reversal after acclimation and training.	[133,134]
Highly trained athletes	Most of them exhibit a markedly high ability to oxidize lipids during exercise but in some sports such as soccer, preferential use of CHO is often observed	[5,59,135,136,137]
Obesity and diabetes	LIPOXmax values markedly shifted to lower power intensities and MFO decreased. After bariatric surgery the LIPOXmax is shifted to lower intensities.	[32,138]
Hypoxia	Exposure to hypoxia did not induce a consistent change in the balance of substrates during exercise compared with normoxia	[139]
Cardiometabolic risk factors	Increased waist circumference and plasma triacylglycerols are associated with impaired lipid oxidation	[140]
Non-alcoholic hepatic steatosis	Inverse correlation between LIPOXmax and the fatty liver index	[141]
Sedentary time, and physical activity time	Sedentary impairs, while physical activity improves, MFO and LIPOXmax. More studies needed.	[142]
Cardiorespiratory fitness (CRF)	MFO and Fatmax are positively correlated with VO_2max_	[143]
Metformin	Increases fat oxidation during exercise and decreases its postexercise rise	[144]
Type 2 diabetes	Lower ability to oxidize lipids when compared to subjects matched for body mass index (difference not found by all authors)	[145,146]
Type 1 diabetes	Lower ability to oxidize lipids	[147,148]
Sleep apnea syndrome	Lower ability to oxidize lipids during exercise. Training improves both apnea index and lipid oxidation during exercise (MFO and LIPOXmax values)	[149]
Fasting	Peak fat oxidation increased in prolonged fasted state and this was highly correlated with plasma free fatty acids concentrations.	[150]
Protein intake	A moderate increase in daily protein intake up to 1 g·kg^−1^·d^−1^ improves the maximal ability to oxidize fat during exercise.	[151]
Anorexia nervosa	Lower ability to oxidize fat during exercise due to the decrease in muscle mass	[152]

**Table 2 nutrients-14-01605-t002:** Studies investigating the effect of LIPOXmax training in overweight. The studies on LIPOXmax training after sleeve gastrectomy are not in this table and are discussed in the text in Section 5.5. It can be seen in this table that most studies (and mostly the randomized trials) are of short duration. The best-designed among these studies have been included in a recent meta-analysis [16], which concludes that this training procedure is undoubtedly an efficient weight-reducing strategy. There are very few long-term studies, and these are ‘real world studies’ representing the follow-up of patients over a long period. Those long-term studies show that even if the weight-lowering effect of LIPOXmax training is rather slow, it is prolonged and allows sustained weight loss over periods as long as 96 months.

Author	Population	Duration of Study	Average Weight Loss
Dumortier et al. (2002) [7]	21 metabolic syndrome	2 months	−2.5 kg
Brandou et al. (2003) [8]	14 obese adolescents	2 months	−3.72 kg
Dumortier et al. (2003) [209]	28 metabolic syndrome	2 months	−2.6 kg
Brandou et al. (2005) [210]	7 obese adolescents	3 months	−5.2 kg
Jean et al. (2006) [211]	28 type 2 diabetics	3 months	−1.3 kg
Ben Ounis et al. (2008) [212]	8 obese adolescents	2 months	−11.5 kg
Fedou et al. (2008) [213]	10 HIV-infected patients under antiretroviral therapy	12 months	−0.92 kg
Bordenave et al. (2008) [86]	11 type 2 diabetics	2 months	No weight change
Venables et al. (2008) [214]	8 obese adults	2 months	−0.2 kg
Ben Ounis et al. (2008) [212]	8 obese boys	2 months	−1.90 kg
Ben Ounis et al. (2008) [215]	6 obese girls	2 months	−1.40 kg
Ben Ounis et al. (2009) [216]	18 obese adolescents	2 months	−6 kg
Ben Ounis et al. (2009) [217]	9 obese adolescents	2 months	−9.5 kg
Romain et al. (2009) [218]	17 psychiatric patients under neuroleptics	3 months	−2.9 kg
Mogensen et al. (2009) [146]	12 type 2 diabetic patients	2.5 months	No weight change
Elloumi et al. (2009) [219]	7 obese adolescent boys	2 months	−1.7 kg
Elloumi et al. (2009) [219]	7 obese adolescents	2 months	−12.3 kg
Maurie et al. (2011) [220]	39 type 2 diabetics	3 months	−2.23 kg
Tan et al. (2016) [221]	29 obese women (20–23)	10 weeks	−4 kg
Besnier et al. (2015) [222]	33 overweight and obese women	5 months	−5 kg
Tan et al. (2016) [223]	15 middle aged women	10 weeks	−3 kg
Tan et al. (2016) [224]	11 boys (9.0 ± 1.0)	10 weeks	−1 kg
Tan et al. (2018) [225]	16 elderly women with T2D	8 weeks	−2.4 kg
Cao et al. (2019) [226]	13 overweight and obese women	10 weeks	−4.6 kg
Zeng et al. (2020) [227]	18 young obese women	12 weeks	NR
Jiang et al. (2020) [228]	13 elderly women with T2D	16 weeks	−2.10 kg
Jiang et al. (2020) [228]	14 elderly men with T2D	16 weeks	−3.3 kg
Hammoudi et al. (2020) [18]	61 obese women	2 years	−6 kg
Brun et al. (2020) [229]	49 obese	6 years	−6.7 kg
Guedjati et al. (2020) [230]	21 middle aged obese women	3 weeks	−2 kg
Bughin et al. (2021) [231]	14 obese followed by telerehabilitation	12 weeks	−0.8 kg
Brun et al. (2022) [33]	10 obese	8 years	−9.1 kg

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
