# Peer review of "Beyond the Calorie Paradigm: Taking into Account in Practice the Balance of Fat and Carbohydrate Oxidation during Exercise?"

_nutrients, 2022, doi:10.3390/nu14081605_

Round 1

Reviewer 1 Report

The present review is extremely long and detailed with over 300 references related to maximal fat oxidation during exercise (LIPOXmax, FATOXmax or FATmax).  It attempts to develop the idea that exercise in the range of maximal fat oxidation is not just a ‘waste of calories’ but a unique range of exercise that has important physiological significance ‘Beyond the calorie paradigm…”.  However, the health benefits of exercise in general, including that at moderate intensity that happen to be within the zone of LIPOXmax are already well known.  The case that LIPOXmax is a unique physiological phenomenon that is regulated precisely, in a similar manner of maximal oxygen uptake (VO2max) is not very convincing to this reviewer for the following reasons.

1) LIPOXmax can occur at a wide range of exercise intensities relative to VO2max. It is recognized that “This level is widely variable among individuals”.  More importantly fat oxidation can be equal at 25% and 65% VO2max (Reference 1).  LIPOXmax does not seem to follow a ‘bell shaped curve’ as discussed, but is rather flat at low intensities and then drops off quickly.  It is not clear if the hand drawn pattern in Figure 2 represents actual data.  The authors do well in relating the idea that “elevation of CHO oxidation leads to diminished fatty acid oxidation during heavy exercise”.  This carbohydrate oxidation seems to regulate fat oxidation as discussed by Sahlin et al.  

2) The sources of fat oxidation during exercise are not even mentioned.  It is well known that ‘intramuscular triglyceride’ (intermyocellular) is important and the source of the increased fat oxidation with training.

3) The significance of the “Crossover Concept” is not immediately clear to this reviewer yet it seems to form the foundation for this review.

4) Carbohydrate feeding in the hours before exercise greatly alter LIPOXmax.  This fact seems to be passed over as is the effect of fasting as well as muscle glycogen depletion.

5) The idea that “training targeted on lipid oxidation is efficient for losing weigh” is interesting and potentially important.  However, most of the studies cited that used moderate intensity exercise did not make comparisons to other intensities.  Therefore the intensity effect can not be distinguished from the simple benefits of exercise per se.

Reviewer 2 Report

The manuscript by Brun and colleagues, entitled “Beyond the calorie paradigm: taking into account in practice the balance of substrate oxidation during exercise?”, reference number Nutrients - 1604729 is an interesting review manuscript, technically and scientifically sound. The English language and style are fine with request for minor spell check. The literature is well cited, fairly covered with recent bibliographic references, including the years of 2019, 2020, 2021 and even 2022. The topic is highly relevant and updated. In fact, the obesity epidemic, with its prohibitive health care and social costs, and the poor success record of available interventions, either behavioral or pharmacological, has prompted the search for novel therapeutic strategies. In this review manuscript, the authors present several lines of evidence supporting the concept that the determination of the balance of substrates during exercise is reproducible, thus providing some new information about metabolic disturbances related to sedentarity and obesity, as well as about adaptations of fuel metabolism in trained athletes. Despite the novelty and interest, I think the manuscript is far too long, very dense and sometimes exhausting for the reader. For this reason, I suggest cutting the manuscript by 25%. All figures (from Figure 1 to 9) should be larger and improved with better resolution to meet the high standards for graphical representations and/or illustrations demanded by Nutrients journal. All in all, the manuscript needs more care in a lot of details (tables, figures, graphical representations).

Apart from these general comments, along I was reading the manuscript, some minor considerations, many of them of structural origin, came up. And here they are, point by point:

 Minor comments: 

  1. In what concerns the title of the manuscript, I suggest including the terms “fat oxidation” and “carbohydrate oxidation”.
  2. I do not believe that Abstract reflects the content of the manuscript and for this reason, I suggest re-formulating the Abstract.
  3. For keywords, please consider the inclusion of CHO oxidation. Moreover, between myokines and myometabokines keywords, select just one.
  4. In line 53, please provide the full meaning of CHO the first time the abbreviation appears.
  5. Regarding methodology, the authors should include a sentence, like this: “The research publications reviewed in the present study were all obtained from Web of Science (Clarivate Analytics, Philadelphia, PA, USA) source from ?? to ?? of the year of ??.
  6. In page 6, line 253, define “healthy obesity”.
  7. In what references concern, be aware of including et al. or colleagues whenever the paper that you are referring to belongs to a team rather than a sole author (see lines 298, 317, 334, 366, 404, 440 and so on throughout the manuscript).
  8. The axis y legend in Figure 3 is not in English. Please, correct this.
  9. In page 9, there are two legend captions for Table 1. Please correct this.
  10. In page 9, please replace “triglycerides” by “triacylglycerols”.
  11. Please provide the full name for FFA abbreviation which is free fatty acids.
  12. In page 11, line 426, please provide the full name for U/S ratio abbreviation.
  13. In lines 464 and 465, I wonder if the values found for MFO and %VO2 max are statistically different?
  14. In page 12, line 477, the font letter is different. Please, correct this.
  15. In page 14, please replace “triglyceride” by “triacylglycerols” (line 561).
  16. IGF-1 should be presented unabbreviated the first time is mentioned rather than as an abbreviation (page 14, line 565).
  17. In line 592, it should be Boisseau and Isacco [116] instead of Boisseau alone.
  18. Please uniformize Tables 1 (page 9) and 2 (page 15).
  19. Please provide BMD unabbreviated when it is presented for the first time (page 17, line 691).
  20. In line 721, Drapier’s study should be followed by its respective reference. Is it [17]?
  21. In line 727, please replace “… -8,49 ± 2,39 kg at the end of the study” by”… -8.49 ± 2.39 kg at the end of the study”.
  22. In Figure 6 caption, provide the bibliographic reference for “… and further updated in January 2022”.
  23. In page 21, lines 891-892, are the values presented significantly different?
  24. Please provide HI and HIIT by full the first time these abbreviations are presented (page 21, line 893).
  25. In page 22, please delete “be” (line 957).
  26. PA what does it mean? (page 26, line 1137).
  27. Please, replace “rôle” by “role” (page 27, line 1196).
  28. Please, replace “The mechanisms of these effects on eating behavior is not clear.” by “The mechanisms of these effects on eating behavior are not clear.” (page 27, line 1198).
  29. Correct the misspelling in page 27, line 1200.
  30. In page 28, from line 1242 to line 1245, if I understood perfectly, the statistics presented is relative to 3618 women and 1640 men [95]… I wonder about the imbalance between the number of men and the number of women. Would you like to comment this observation?
  31. In my opinion, the Concluding remarks are very well presented contrarily to the Abstract section (page 29).
  32. The literature is well cited, fairly covered with recent bibliographic references, including the years of 2019, 2020, 2021 and even 2022.
  33. All figures (from Figure 1 to 9) should be larger and improved with better resolution to meet the high standards for graphical representations and/or illustrations demanded by Nutrients. The figures are the least achieved aspect of the manuscript.

Round 2

Reviewer 1 Report

The data upon which these theories are developed continues to be a mystery.  The graphs that supposedly represent fat metabolism misleading or often wrong.  They appear to be hand drawn based upon the authors loose interpretation of the literature.  For example, they display fat oxidation as being higher that lipolysis during some intensities of exercise.  That is impossible and not what is documented in the literature.  This is because the stores of plasma FFA are very small and there is no buffer as the turnover is high.  My point is that this review is not based on careful data from the literature but instead based on loose theories.

Reviewer 2 Report

This second version of the manuscript by Brun J.-F. and colleagues, now entitled “Beyond the calorie paradigm: taking into account in practice the balance of fat and carbohydrate oxidation during exercise?”, reference number Nutrients - 1604729 is very much improved. The authors addressed and very well all the concerns and corrections suggested by this reviewer. As so, at this stage, I have only a small amount of minor revisions and few comments that I would like to see corrected.

And here they are, point by point:

 Minor comments: 

  1. In page 1, line 12, the word “Recent” is in bold. Please, correct this.
  2. Regarding the Keywords, “CHO oxidation” should be replaced by “carbohydrate oxidation”.
  3. In what references concern, be aware of including et al. or colleagues whenever the paper that you are referring to belongs to a team rather than a sole author (throughout the manuscript). This has not been corrected.
  4. In page 10, line 386, please replace “This increase in plasma is free fatty acids was related to the transient carbohydrate deficit after exercise [150].” by “This increase in plasma free fatty acids was related to the transient carbohydrate deficit after exercise [150].”
  5. In page 13, line 499, please replace “Growth hormone” by “growth hormone”.
  6. In page 13, line 503, please replace “Insulin-like growth factor” by “insulin-like growth factor”. The same in Abstract (page 1, line 25).
  7. In page 25, line 1015, please replace “Ghrelin” by “ghrelin”.
